# Subspheroids in the lithic assemblage of Barranco León (Spain): Recognizing the late Oldowan in Europe

**Stefania Titton**[1,2]☯*, **Deborah Barsky**[1,2]☯, **Amèlia Bargalló**[3]☯, **Alexia Serrano-Ramos**[4,5]☯, **Josep Maria Vergès**[1,2]‡, **Isidro Toro-Moyano**[6]‡, **Robert Sala-Ramos**[1,2]‡, **José García Solano**[4]‡, **Juan Manuel Jimenez Arenas**[4,7,8]‡

**1** Institut Català de Paleoecologia Humana i Evolució Social (IPHES), Tarragona, Spain, **2** Area de Prehistoria, Universitat Rovira i Virgili (URV), Tarragona, Spain, **3** Institute of Archaeology, University College, London, United Kingdom, **4** Department of Prehistory and Archaeology, University of Granada, Granada, Spain, **5** Laboratorio 3D de Modelización Arqueológica, Department of Prehistory and Archaeology, University of Granada, Granada, Spain, **6** Archaeological and Ethnological Museum of Granada, Granada, Spain, **7** Instituto Universitario de la Paz y los Conflictos, Universidad de Granada, Granada, Spain, **8** Department of Anthropology, University of Zürich, Zürich, Switzerland

☯ These authors contributed equally to this work.
‡ These authors also contributed equally to this work.
* stefania.titton1@gmail.com

**Data Availability Statement:** All relevant data are within the manuscript and its Supporting Information files. Lithic material inventories and basic analyses may be consulted in the Field

## Abstract

The lithic assemblage of Barranco León (BL), attributed to the Oldowan techno-complex, contributes valuable information to reconstruct behavioral patterning of the first hominins to disperse into Western Europe. This archaic stone tool assemblage comprises two, very different groups of tools, made from distinct raw materials. On the one hand, a small-sized toolkit knapped from Jurassic flint, comprising intensively exploited cores and small-sized flakes and fragments and, on the other hand, a large-sized limestone toolkit that is mainly linked to percussive activities. In recent years, the limestone macro-tools have been the center of particular attention, leading to a re-evaluation of their role in the assemblage. Main results bring to light strict hominin selective processes, mainly concerning the quality of the limestone and the morphology of the cobbles, in relation to their use-patterning. In addition to the variety of traces of percussion identified on the limestone tools, recurrences have recently been documented in their positioning and in the morphology of the active surfaces. Coupled with experimental work, this data has contributed to formulating hypothesis about the range of uses for these tools, beyond stone knapping and butchery, for activities such as: wood-working or tendon and meat tenderizing. The abundance of hammerstones, as well as the presence of heavy-duty scrapers, are special features recognized for the limestone component of the Barranco León assemblage. This paper presents, for the first time, another characteristic of the assemblage: the presence of polyhedral and, especially, subspheroid morphologies, virtually unknown in the European context for this timeframe. We present an analysis of these tools, combining qualitative evaluation of the raw materials, diacritical study, 3D geometric morphometric analysis of facet angles and an evaluation of the type and position of percussive traces; opening up the discussion of the late Oldowan beyond the African context.

Season Reports that the Orce Research Team delivers yearly to Junta de Andalucia, which are freely available by request. The studied material is preserved in the Archaeological and Ethnographical Museum of Granada (Andalucia, Spain).

**Funding:** This research has been funded by the Junta de Andalucía, Consejeria de Educacion, Cultura y Deporte: Orce Research Project "Primeras ocupaciones humanas y contexto paleoecológico a partir de los depósitos pliopleistocenos de la cuenca Guadix-Baza: zona arqueológica de la Cuenca de Orce (Granada, España), 2017–2020"; "Presencia humana y contexto paleoecológico en la cuenca continental de Guadix-Baza. Estudio e interpretación a partir de los depósitos Plio-Pleistocénicos de Orce. Granada. España" B120489SV18BC, 2012-16; "Primeras ocupaciones humanas del Pleistoceno inferior de la cuenca de Guadix-Baza (Granada, España)" B090678SVI8BC, 2009-11; MICINN (no feder) "Estudio de las dispersiones faunísticas y humanas durante el Pleistoceno inferior en la cuenca mediterránea.", CGL2016-80975-P, 2017-19; the Spanish government Ministerio de Ciencia, Innovación y Universidades (MICINN-FEDER) code CGL2016-80975-P, and the Generalitat de Catalunya Research Group 2017SGR 859. "Comportamiento ecosocial de los homínidos de la Sierra de Atapuerca durante el Cuaternario V", MICINN-FEDER PGC2018-093925-B-C32 and the Generalitat de Catalunya, AGAUR agency, SGR 859 and SGR 1040. Gerda Henkel Foundation (AZ 32/V/19, Lower Paleolithic Spheroids Project (LPSP) is assuring continuity in this line of research. ST is beneficiary of the Provincia Autonoma di Bolzano (Italy) post-master scholarship. AB has been funded from the European Union's Horizon 2020 research and innovation programme under the Marie Sklodowska-Curie Action grant agreement PREKARN nº702584. The research of DB, JMV, & RSR is funded by CERCA Programme/Generalitat de Catalunya. JMJA belongs to the Research Group HUM-607.

**Competing interests:** The authors have declared that no competing interests exist.

## Introduction

Archaic stone tools are not only related to the genus *Homo* [1], as evidenced, for example, at the Lomekwi 3 site, where the lithic assemblage (3.3 Ma, West Turkana, Kenya) predates its emergence [2, 3]. While *it is not currently possible to positively identify the creators of the earliest stone tools at Gona* [4: 627], fossils attributed to *Australopithecus garhi* found in the Hata Member of the Bouri Formation (Middle Awash) show equivalent age to the EG and OGS sites (2.6–2.5 Ma., Ethiopia) [5, 6, 7, 8, 9]. Meanwhile, the newly discovered industry of Bokol Dora 1 (BD 1) (Ethiopia), dated from 2.61 to 2.58 Ma [10], is situated near the Ledi Geraru site, where the most ancient remains of the genus of *Homo* have been identified (2.8 Ma., [11]). It is noteworthy that, from a technological point of view, the older Lomekwian tools present significant differences from those attributed to the Oldowan techno-complexe [10]. It therefore appears that more than one species of hominin–not all of the genus *Homo*—began to rely ever more significantly upon technologies, in an adaptive shift to limit constraints posed by the environment through object mediation. This change generated a process in which our hominin ancestors would come to distinguish themselves from other primates by learning a comparatively high degree of technological skills. The early stages of this process are reflected materially in the "Oldowan Industrial Complex" (OIC), a term coined in 1936 by Luis Leakey [12] and that came into general use from the 1970's [7, 13, 14, 15, 16, 17, 18] (or Mode 1, [19]). In its early context, the OIC grouped together lithic assemblages from early African sites [20] with strong similarities, characterized by homogeneity and variability as pointed out by Barsky [21] and Carbonell and colleagues [22]. The roots of this techno-complex are distinguished by knapping processes applied to rock matrices with a hard hammerstone, using basic operational schemes to obtain flakes [23, 24]. This behavior has been shown to indicate relatively high cognitive abilities and *technological competence* of the hominins [25, 26, 27]. While technical variability is observed within the OIC, its foundational features are largely uniform: small, non-retouched flakes, unidirectional or orthogonal core types, generally accompanied by a larger-sized pounding toolkit. However, Oldowan variability as described by Carbonell and colleagues [22], and Barsky [21], is attributable precisely to these same factors; most significantly, the different morphologies obtained using the unidirectional and orthogonal core reduction methods. Final core morphologies will vary in accordance to the raw materials, the length of knapping episodes and the cobble's formal attributes. For classification purposes, these forms are attributed different denominations, while they in fact represent different stages in the application of these simple knapping systems (with little or no platform preparation). Sometimes, new reduction systems did occur within the OIC [28], allowing hominins to move beyond the mechanics of the unifacial and unidirectional strategies, and to explore multidirectional core management strategies [29].

In some cases, the assemblages also contain polyhedrons, subspheroids and spheroids, as well as scarce heavy-duty and light-duty tools [13]. The morphological and quantitative variability of Oldowan tool forms was initially described by Mary Leakey, who created subdivisions within the complex, based on Olduvai Gorge sites (Bed I, Lower Bed II [13]). Leakey [13], thus describes the assemblages as belonging to the *Classic Oldowan* (O) or to the *Developed Oldowan* (DO), in accordance to typological components, with the latter referring to assemblages displaying higher morpho-technological variability. Leakey [13] further divided the DO into three categories (A, B and C), based on chronological criteria as well as on quantitative variability of the different morphotypes recognized in the assemblages (for an overview of the debate on evolving stages within the Oldowan see: [8, 30, 31, 32, 33]). Importantly, polyhedrons, spheroids and subspheroids (PSSB, [34]) were among the tool forms that Leakey described as indicative of cultural variability within the DO [13]. These morphologies have

been described and categorized into different typological categories, for example, by Klein-dienst [35] and Leakey [13]. These tool categories, still valid today [36], share a volumetric structure organized around a central point, summarized as follows:

- **Polyhedrons.** Objects characterized by angular contours, displaying three or more, usually intersecting, working edges.

- **Subspheroids.** More or less rounded items, that often display cortical surfaces and protruding negative crests (facets).

- **Spheroids.** Rounded objects with negatives presenting facetted surfaces covering their entire surface area.

- **Bolas.** Stone balls characterized by the absence of protruding negative crests, in which phases of continuous "*piquetage*" have effaced surface angularities, giving them a perfectly spherical shape.

Based on these descriptions, the categories comprising the PSSB are distinguished from each other by direct observation; basically by recognizing deviances from the regular morphological-volumetric aspect of a sphere, developed around a central point (center of mass), which confers them a more or less rounded shape.

Over the years, researchers have tried to determine what the function(s) of these stone balls might have been (for a summary of functional interpretations: [36, 37]), but this conundrum remains unresolved today [34] and major questions continue to revolve around these tools. Some authors, for example, have argued that spheroids and their derivative forms were intentionally manufactured in accordance to a pre-conceived template, and that they are the result of a reasoned organizational process that can be segmented into a single operational chain [34]; or even that they were shaped from an independent operating scheme [38]. This supports the hypothesis that these rounded tools were 'configured' to fit specific functional needs [39, 40]. Others have interpreted spheroid morphologies as the outcome of a *chaîne operatoire*, preferring to qualify them as cores [37, 41, 42, 43]. Still others have proposed that spheroids can be accidentally produced from recurrent percussive activity [44, 45], or even that their morphology could be the haphazard result of a poorly skilled knapper [46, 47]. The hypothesis that spheroid morphologies could represent projectiles has also been explored based on studies of mass distribution and dynamic analysis [48, 49, 50].

This paper presents new findings from the late Lower Pleistocene Oldowan site of Barranco León (BL) (Andalusia, Spain), following the recent identification of some subspheroid morphologies in the assemblage, in order to test their possible assimilation with the morpho-technological variability of the PSSB, as described above. This paper presents data from a detailed analysis of these tools, contributing new information to discussions about the European Oldowan by taking a step forward towards understanding these tool morphologies. We also present a methodology designed specifically for the analysis of spherical morphotypes, combining different approaches:

1. **Raw materials qualitative evaluation** allows to discriminate between the different types of limestone and to identify selective criteria guiding the hominins in their choice of cobbles. In addition, we evaluate the role of taphonomic alterations that, in some cases, are restrictive to obtaining results.

2. **Diacritical study** is applied to better understand the technical processes involved in the management of spherical morphologies, and to identify, whenever possible, systematic technological strategies.

3. **Differential morphometric analysis** (manual and computer-sourced) was undertaken to register the dimensional features of the spherical tools and their surface negatives, and to identify possible recurrences. Given the relative complexity of the tools in terms of number of facets and sphericity, we complement these classical measurements with a digital analysis, focusing on the amplitude of the angles separating the facets, as a key factor contributing directly to the overall spherical morphology of the tools. We apply statistical analysis to 3D computational reproductions of the tools, in order to distinguish their possible attribution to one of the morphological groups of the PSSB, thus offering new criteria with which to move beyond simple (subjective) visual selection [13, 34, 35].

4. **Analysis of percussion marks** is grounded in the methodology we developed in previous studies of limestone macro tools [51, 52], linking the type and overall morphology of the traces to the volumetric features of the original cobbles.

Our study presents a detailed account of this methodology applied to the BL spherical tools, in the aim of understanding whether they were accidentally produced during percussive activities, or, alternatively, if they result from intentional configuration involving a true operative scheme. The results from this study are discussed in the framework of formulating hypotheses about the cultural significance of spheroid-type tools at BL, and, more largely, in the late Oldowan in Europe.

## 2. Context of the BL site

The archeo-paleontological site of BL is located in the distal lacustrine sector of the Guadix-Baza Basin (Orce, Andalucia, Spain). This intra-mountain basin is situated in the contact area between the External and Internal Zones of the Baetic Ranges. During the Lower Pleistocene, the Guadix-Baza Basin contained a large, endorrheic lake system, situated in the Baza sub-basin. The Baza lake was fed by three main sources: the Fardes River, flowing from SW to NE, small rivers coming from adjacent mountain ranges, and hot springs [53, 54, 55, 56, 57]. Towards the end of the Middle Pleistocene [58] or during the Upper Pleistocene [59, 60], the later catchment of lake waters by the drainage network of the Guadalquivir River dramatically changed the dynamic of the lake basin, which became exoreic. Due to subsequent headwater erosion, the Baza sub-basin became punctuated by deep gorges and ravines (Spanish *barrancos*) (Fig 1A), exposing an exceptionally rich archeo-paleontological record of large and small vertebrate fossils, as well as stone tools bearing witness to the presence of hominins in the area since the earliest European Oldowan [61, 62]. Specifically, the BL (Fig 1B) and Fuente Nueva 3 (FN3) sites [63], located near the town of Orce, are situated in a shallow lacustrine facies of the Upper Member of the Baza Formation [64], close to the lake margin [54]. Thanks to their rich lithic [51, 52, 65, 66, 67] and faunal assemblages [68, 69, 70, 71, 72, 73, 74, 75, 76, 77, 78], both of these sites have yielded important information about the lifeways of the oldest hominin groups present outside of Africa some 1.4 and 1.2 Ma, respectively [79, 80, 81, 82, 83, 84, 85]. The hominins presence is now well-attested in Europe from this timeframe onwards (Fig 1C). The BL site has yielded a deciduous molar that represents the oldest hominin remain discovered so far in Europe [86].

The age of BL level D, established at 1.4 Ma, was obtained by combined relative and absolute dating methods, including: micro and macro mammal biochronology, magnetostratigraphy, U-series and ESR dating [53, 82, 83, 87, 88, 89, 90, 91, 92, 93]. Recently, level D has been subdivided into two sublevels (D1 and D2) (Fig 1B), corresponding to two distinct depositional events [57, 94]. Although they share the same sandy sedimentary matrix, level D1 materializes a high-energy, rapid water transport of gravels and cobbles, while level D2 is representative of a more gradual *in situ* sedimentary deposition.

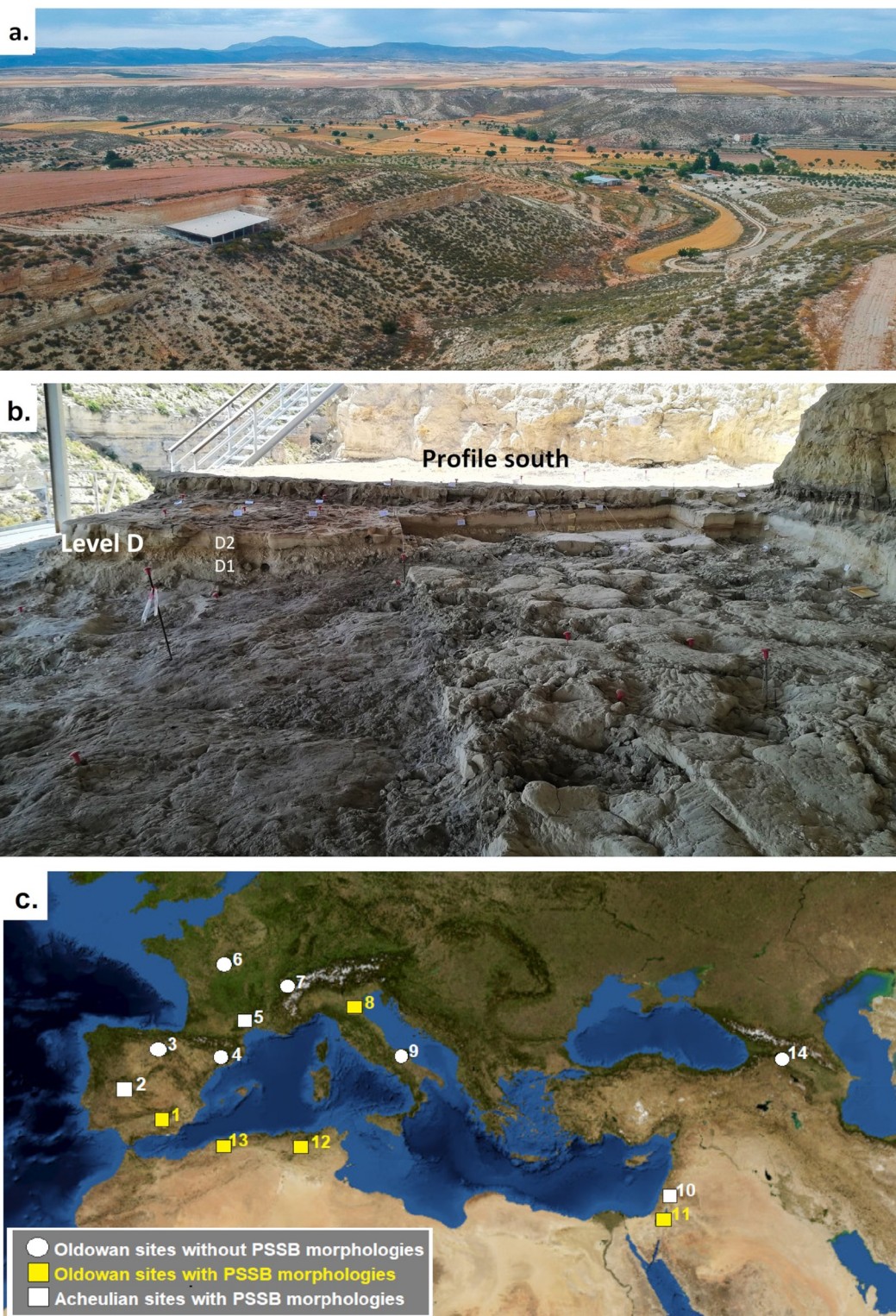

**Fig 1. Geographic situation of the BL site. 1a.** View of the current landscape configuration in which the BL site is located (Photo: F.L. Chmiel). The BL site (on the left) was near to the edge of the paleo-lake, and is situated today on the edge of a ravine with a N-S direction. **1b.** General view of the BL site. **1c.** Geographic position of some circum-Mediterranean sites in Eurasia that have yielded Oldowan lithic assemblages with (yellow dot and underlined in the text) or without PSSB morphologies indicated by bibliography. (USGS National Map Viewer).—Circum-Mediterranean Oldowan sites in Eurasia

with and without polyhedral/subspheroid morphologies: **(1.)** BL and FN 3 (1.4 and 1.2 Ma., Andalusia, Spain); **(3.)** Levels TD6 Gran Dolina (0.8–0.9 Ma.) [95, 96, 97], and TE9 at Sima del Elefante, Atapuerca (1.2 Ma, Castilla y León, Spain) [98, 99]; **(4.)** Vallparadís (*ca*. 0.98 Ma, Catalonia, Spain) [83, 100]; **(6.)** Pont de Lavaud (1,1 Ma., Indre, France) [99, 101]; **(7.)** Le Vallonnet (*ca*. 1.2 Ma., Roquebrune-Cap-Martin, France) [102, 103]; **(8.)** Ca' Belvedere di Monte Poggiolo (*ca*. 1 Ma., Emilia Romagna, Italy) [104, 105, 106, 107]; **(9.)** P13 locality of Pirro Nord (1.6–1.3 Ma., Puglia, Italy) [108, 109, 110, 111]; **(11.)** Bizat Ruhama (Israel, 1.6–1.2 Ma) [112]; **(12.)** In North Africa, Ain Hanech (*ca*. 1.8 Ma, Algeria) [42, 113]; **(13.)** and Ain Boucherit (ca. 1.9 and 2.4 Ma, Algeria) [114]; **(14.)** Dmanisi (1.85–1.78 Ma., Georgia) [115, 116, 117].—Circum-Mediterranean Acheulian sites in Eurasia with polyhedral/subspheroidal tool morphologies: **(2.)** Santa Ana cave (ca. 117–183 Ka. BP, Extremadura, Spain) [118]; **(5.)** US4 Bois-de-Riquet site (ca. 0.8 Ma.), Lezignan-la-Cèbe, Herault, France) [119]; **(10.)** and 'Ubeidiya (1.6–1.2 Ma, Israel) [120, 121, 122].

The Baza saline lake and its surrounding fresh-water sources and hot springs certainly provided favorable conditions for numerous animal species, including early hominins. This scenario is coherent with paleoclimatic data, which indicates warmer winters and more humid conditions than today [76, 77].

## 3. Materials and methods

Following the latest revision of the lithic assemblage recovered from the BL site (2018), the collection comprises a total of 2 464 pieces, all collected during systematic excavations taking place continuously from 1985 (Table 1). The majority of the assemblage is composed of debitage, knapped from Jurassic-age flint (1 562 pieces 72.7%). The limestone cobble-toolkit comprises 581 pieces (27% of the assemblage), including abundant percussive materials [51, 52]. Finally, the collection also contains 313 non-modified cobbles that have been collected to serve for comparison with the sedimentary context (flint pebbles = 58; limestone cobbles = 249; cobbles in other materials = 6). While this material forms a part of the natural context, the presence of manuports is not excluded. There are a few pieces knapped from crystalline material whose precise petrographical nature remains to be determined (8 pieces, 0.3% of the assemblage).

The BL lithic assemblage is characterized by numerous medium to very small-sized chipped fragments and flakes of flint (1093 pieces 50.8% of the assemblage), and limestone cobbles (247 pieces 11.5% of the assemblage). The preferential use of flint or limestone for different purposes is clearly evidenced by their relative proportions within the structural categories of the assemblage (Table 1). Flint is the most abundant raw material for the debitage, and the

**Table 1. Frequency of the different structural categories composing the lithic assemblage from BL.**

| Structural category | Limestone | | Jurassic flint | | Others | | Total | |
|---|---|---|---|---|---|---|---|---|
| | N | % | N | % | N | % | N | % |
| Flakes and broken flakes | 59 | 2,7% | 421 | 19,6% | 2 | 0,09% | 482 | **22,40%** |
| Debris < 5 cm and flake fragments | 247 | 11,5% | 1092 | 50,8% | 5 | 0,23% | 1344 | **62,50%** |
| Debris > 5 cm (fragments and broken cobbles) | 147 | 6,9% | - | - | - | - | 147 | **6,90%** |
| Cores | 65 | 3,0% | 49 | 2,3% | 1 | 0,05% | 115 | **5,30%** |
| Polyhedron, spheroid and subspheroid morphotypes | 5 | 0,2% | - | - | - | - | 5 | **0,20%** |
| Heavy-duty scrapers | 7 | 0,3% | | | | | 7 | **0,30%** |
| Percussion tools (active and passive) | 51 | 2,4% | - | - | - | - | 51 | **2,40%** |
| **Total** | **581** | **27,0%** | **1562** | **72,7%** | **8** | **0,3%** | **2151** | **100%** |
| (*) Pebbles and cobbles without anthropic traces | 249 | | 58 | | 6 | | 313 | |
| Total | 830 | | 1620 | | 14 | | 2464 | |

(*): Sedimentary materials collected for comparative purposes.

presence of flakes from different phases of core reduction reflects on-site knapping of this material (flakes and debris with various degrees of residual cortex, abundant tiny fragments and some cores).

Flint cores are scarce (3.1% of the 1 562 pieces in flint) and intensively reduced compared to the limestone (11.2% of the 581 pieces in limestone). This greater intensity of reduction can be explained by the categorical differences in the assemblage concerning these two rock types: flint was used for obtaining small flakes, while limestone served mainly for percussive activities. It can also be explained by the relative scarcity of flint compared with limestone in the immediate vicinity of the site. Flint was collected in detrital position as small nodules, cobbles or plates, and the cores are very small. Contrastingly, the limestone was collected as cobbles of varying sizes and shapes [51, 52]; limestone cores are bigger than the flint ones and they often present reserved cortical surfaces. Bipolar-on-anvil stone reduction played an important role in the exploitation of both of these rock types; while freehand hard hammer methods are also recognized as significant [63]. Core reduction strategies, achieved by both of these methods, are described [63, 65] as recurrent and unifacial or orthogonal.

On-site knapping of the flint is further attested by the comparatively high frequency of flakes and small fragments (27.9% and 69.9% of the flint assemblage, respectively). Contrastingly, the limestone was used for different purposes; mainly linked to percussive activities. However, the limestone was also used for flake production, as is attested by the relatively numerous limestone cores (11.1% of the 581 pieces in limestone and 56.5% of all the cores), as well as small and large fragments (42.5% and 25.3% of the limestone, respectively). Limestone cores are: unifacial (33.8% of the limestone cores) (types: unifacial unipolar, unifacial semi-peripheral and unifacial peripheral, unifacial centripetal, unifacial bipolar); bifacial (50.8% of the limestone cores) (types: bifacial bipolar, bifacial orthogonal with 2 and 3 directions of removals, bifacial multipolar); and multifacial (15.4% of the limestone cores) (type: multifacial multipolar).

After reviewing the entire lithic collection, five limestone pieces (0.2% of the whole collection) were classified by their morphological characteristics and special technological features as attributable to the PSSB group as defined by Kleindienst [35], Leakey [13] and more recently by Tixier & Roche [34]. We compare the pieces that share the multifacial and multidirectional management strategies, including those with semi-rounded/rounded morphologies (the PSSB) (Fig 2), using diacritical analysis. In order to evaluate whether or not the PSSB can be isolated with respect to the cores, morphometric and statistical analyses were carried out. To overcome the difficulties relating to surface preservation of these pieces and to recognize and verify their special features and assess the direction of their removal negatives, we decided to compare them with 2/10 of the multifacial multipolar cores. In fact, only these two cores presented surface conservation sufficient to realize a diacritical study (visible impact points and removal directions). In order to better understand the structure of the objects in relation to their removal negatives, we reconstructed the different phases of the production process that led up to their subspheroid morphology, following the *chaîne opératoire* hypothesis for spheroid production [34, 38, 123, 124, 125]. The diacritical study [126, 127, 128, 129] was carried out to analyze the position of the impact points and the direction and morphology of the negatives visible on the cores and 'potential' subspheroid morphotypes.

The methodology was developed in four stages:

**Stage 1. <u>Qualitative evaluation</u>** of the limestone raw material was performed by macroscopic analysis: visual observation and tactile perception of the limestone raw material characteristics. Four stages of alteration were identified based on the rounding of the crests separating the negatives and the overall surface condition of the tools (Fig 3). These variables

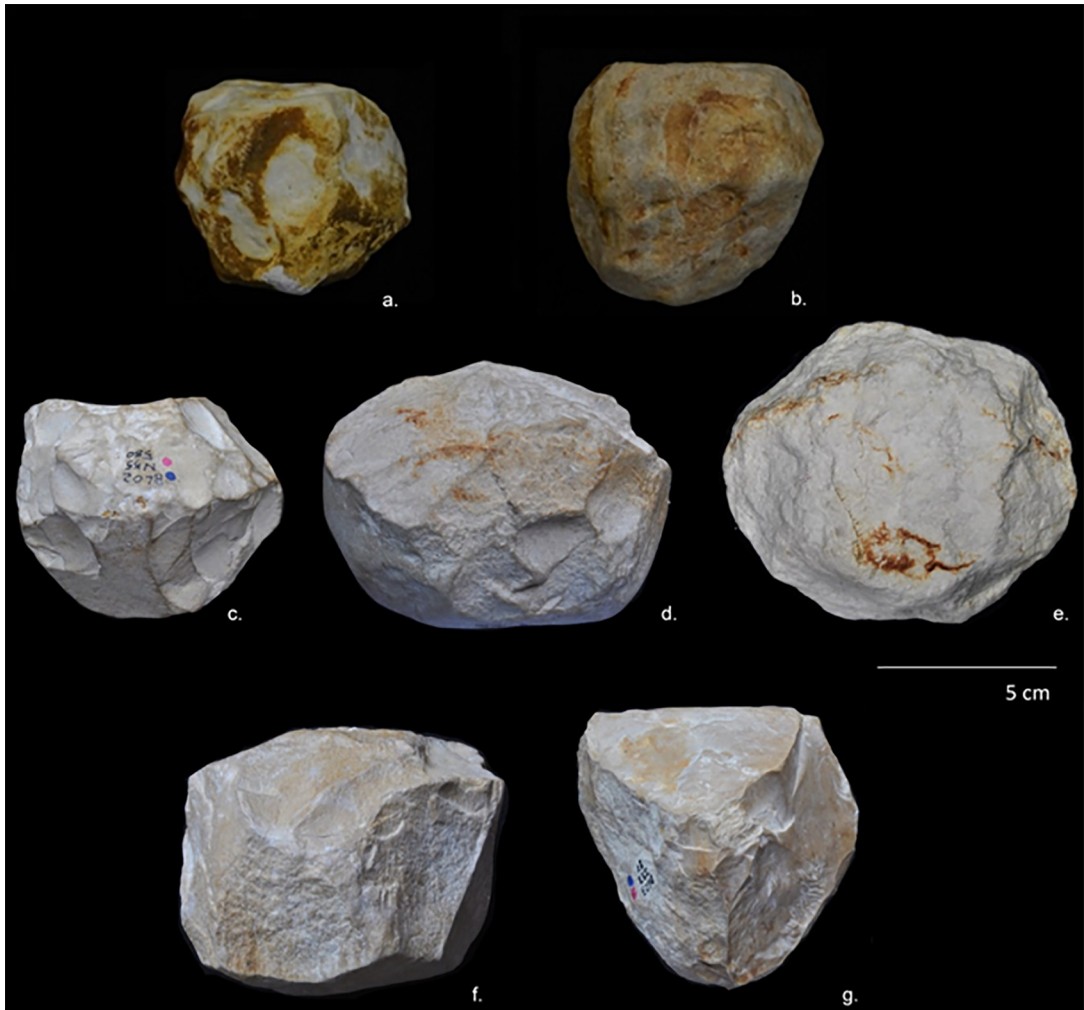

**Fig 2. Polyhedral and subspheroid morphotypes from the BL site (a-e), and multidirectional cores selected for the diacritical analysis (f-g).** Specimen: Tool a: BL.2014.D2.G49.114; Tool b: BL.2014.D2.M49.49; Tool c: BL.2002.D1.N55.580; Tool d: BL.2002.D1.M54.500; Tool e: BL.2018.D1.N41.1; Tool f: BL.2002.D1.J52.69; Tool g: BL.2003.D1.I53.27. Stored at the Archaeological and Ethnological Museum of Granada (Spain). All necessary permits were obtained for the described study, which complied with all relevant regulations.

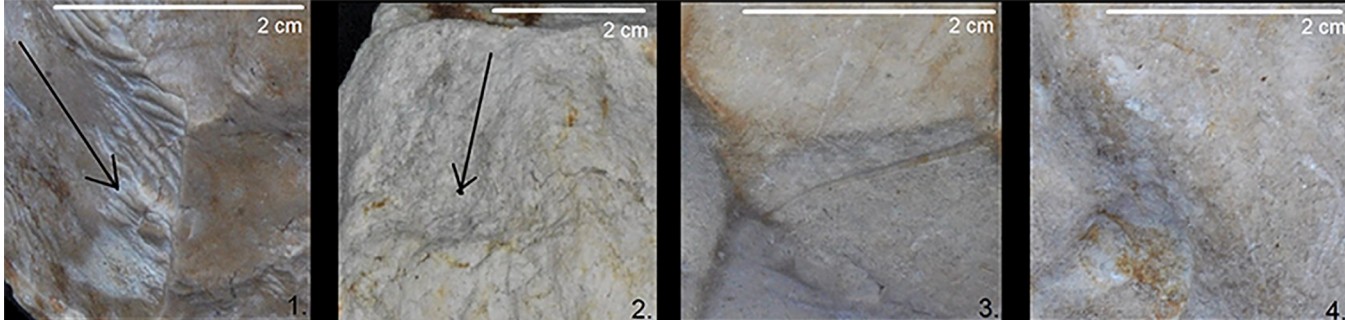

**Fig 3. Different degrees of alteration.** (left) Level 1: Not altered; (center left) Level 2: Slightly altered; (center right) Level 3: Altered; (right) Level 4: Very altered (Photos: S. Titton).

are affected by the composition of the limestone (degree of silification), as well as on taphonomic alteration (mainly water circulation);

- Level 1: Not altered. Clear reading of the direction of the negatives, with fresh markers allowing recognition of superimposed negatives and knapping directions thanks to visibility of surface striations and undulations;

- Level 2: Slightly altered. Surface alteration does not entirely hinder the diacritical reading of the negatives: it is possible to recognize negative superimpositions, knapping directions and impact points;

- Level 3: Altered. The alteration conceals the distinctive characteristics of the removal negative surfaces by smoothing their structural features. Although it is not possible to recognize the order of superimposition, removal negative concavities generally allow to distinguish the direction of the extractions;

- Level 4: Very altered. The degree of alteration does not allow to read the direction of the negatives, or to evaluate their integrity.

**Stage 2) Diacritical study.** Diacritical analysis applied to the study of stone tools aims to reconstruct the main phases of reduction involved in their manufacture (in this case, obtaining PSSB tools).

Each tool was drawn by overlapping the profiles with their photos (Fig 4A and 4B), and checked by direct observation of the pieces. General topographic features and the direction of the extractions were recorded whenever possible [129] (Fig 4C).

Depending on the level of alteration of the raw material (Fig 3), each readable negative and its source-direction was identified and numerically sorted based on the recognition of the overlap separating the removals. The rupture of the morphologies and of the *theoretical* volume of the negatives [126] were recognized as determinant criteria for defining superimposition. Furthermore, the order of the extractions was established by observing the morphology of the negative edges and the intersections between them [126].

In the diacritical drawings presented here, the number pertaining to the order of the extractions is repeated when it is impossible to determine the sequentiality between two or more negatives as follows:

- Negatives with the same number and the symbol (*)

The symbol (*) indicates that there is another negative with the same number. This occurs when: 1) the negatives are not continuous and their disposition does not allow to identify diachrony; 2) when negative surfaces are eroded; when the negatives are separated by a cortical surface or transected by another negative.

They are identified:

- on the same profile (Fig 5, red circle)

- on different profiles (Fig 5, rectangle).

When situated on the same profile, measurements of the negatives were carried out from left to right.

- Negatives with the same number and without the symbol (*)

Negatives illustrated with the same number but without the symbol (*) indicate the same removal viewed from different angles. This allows for easier interpretation of the volumetric characteristics of each piece and its extractions, by viewing the different profiles (Fig 5, black line).

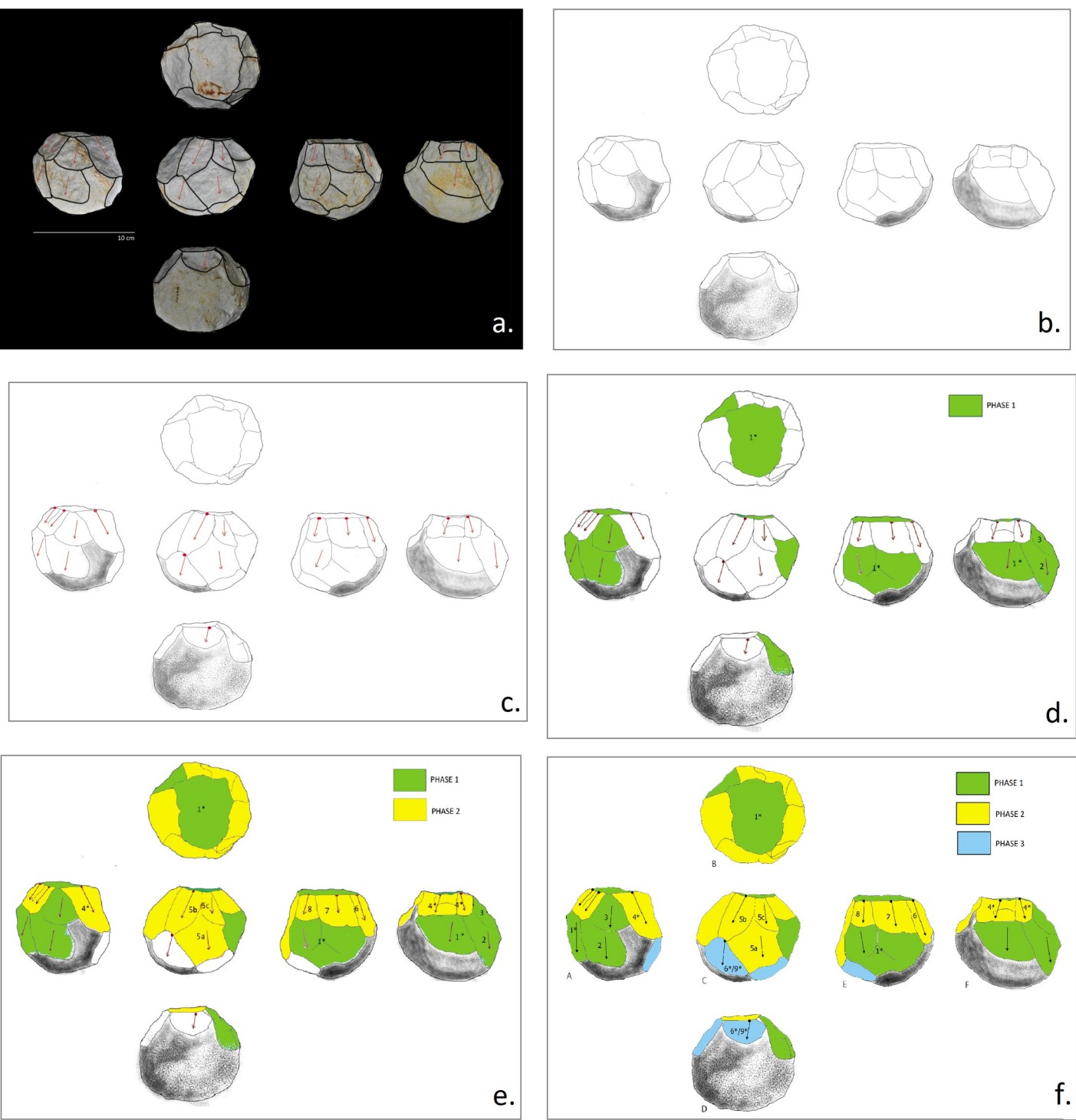

**Fig 4. Graphic representation of the methodology used for the diacritical drawings of the polyhedral/subspheroid lithic tools from BL.** Impact points and negative directions are indicated and the different reduction phases are highlighted by different colors.

In parallel, and consequential to the diacritical reading, different phases of reduction are highlighted for each piece (Fig 4D, Fig 4E, and Fig 4F). In addition, different phases of

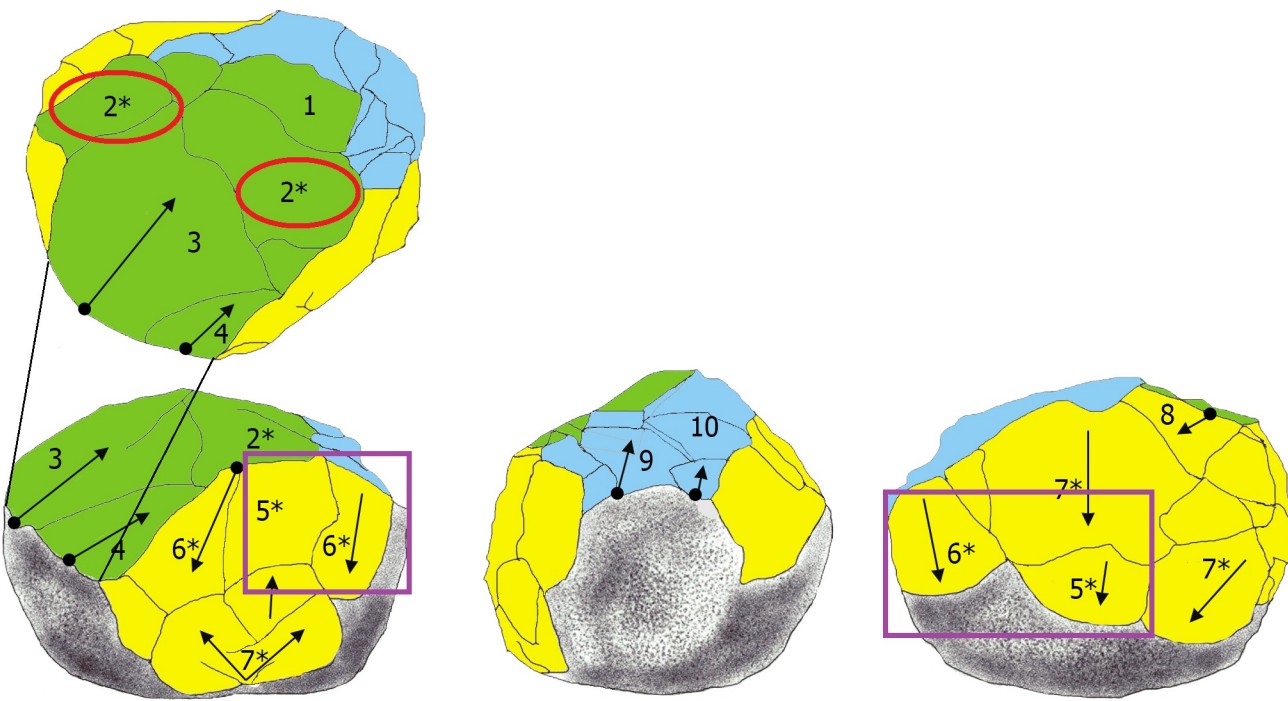

**Fig 5. Graphic representation of the diacritical methodology used to interpret the polyhedral/subspheroid lithic tools from BL.** The different colors of the negatives represent the different phases of reduction.

retouching and use (traces of percussion) are indicated whenever possible, indicating their relationship with respect to the sequence.

The aim of this stage of diacritical analysis is to compare the organizational systems used to obtain polyhedron core forms, sub-spheroid morphotypes and multidirectional cores at BL, in order to identify any similarities that could exist in the management of the extractions (operative schemes). Additionally, this methodology allows to discern whether these rounded forms resulted from an unorganized series of blows, or if they reflect intentional configuration.

### Stage 3.) <u>Morphometric analysis</u>

As a next step, three-dimensional images of the objects were generated (3D images: https://sketchfab.com/TITTONETAL2019) to obtain precise measurements of the angles separating the negatives and to evaluate possible discrimination based on this key aspect lending the spheroid morphology to the pieces.

The maximum measurements for each tool (maximum length, maximum width (calculated perpendicularly to the length) and maximum thickness (measured perpendicular to the width). The size of the facets (surfaces without readable extraction features) and negatives and the amplitude of the angles have been recorded. The measurements were carried out in two distinct stages.

Stage 3.1 a) Basic measurements: The maximum length, width and thickness were measured in mm; b) Measurements of negatives and facets: Whenever possible, the length of the negatives was measured manually to discriminate by direct observation, between whole versus incomplete negatives (length was measured from the point of impact to the distal extremity; and width is perpendicular to the latter).

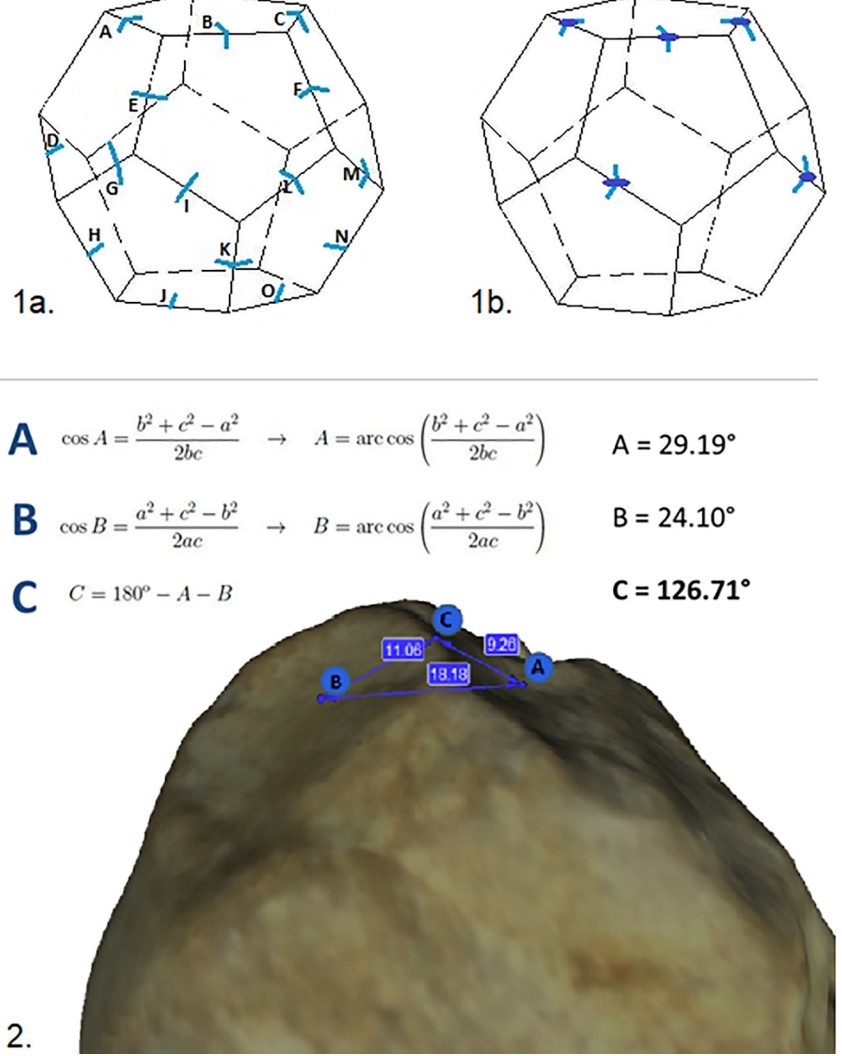

**Fig 6. Representative images of angles detection.** 1a) Example of angles measured between the facets; 1b) In order to determine the percussion angle, only the corners displaying an impact point are taken into account; 2) Example of angle detection used for trigonometric calculation.

Stage 3.2: Measurement of the angles: The tools were digitalized using a structured light Artec 2000 scanner, which allows a maximum error of 0.5 millimeters. Artec Studio X Professional, which incorporates measuring tools. Each corner of the distinct facets was distinguished by a letter (Fig 6.1a). In addition, to determine the percussion angle for each removal, we selected those with an impact point and measured the mean of these angles separately (Fig 6.1b).

The Artec 2000 linear measuring tool allows to calculate distances between selected points and was used to obtain the dimensions of each of the negatives. By selecting three points and with known measurements of the three sides of the resulting "triangle" (points A, B, C), trigonometric calculation permits precise evaluation of the angles (C). As the points display their coordinates, it is possible to set the third point above the first point, thus creating a triangle (Fig 6.2).

Stage 3.3 Evaluation of morphological discrimination based on the amplitude of the angles.

The variability of the angles separating the negatives and creating the facets was calculated for each tool, in order to bring to light possible discriminatory factors distinguishing between the morphotypes. Results from this analysis serve to identify volumetric relationships based on the amplitude of the inter-facet angles measured for each piece.

In order to differentiate between data sets, the grouping range was determined in accordance to resuts obtained by our statistic analysis [130].

Lubischew's test [131] was used as an objective means by which to assess morphological discrimination, separating groups within the sample based on the different tendencies of their components. Lubischew's test is a parametric statistical tool, based on the Student's Test, which enables us to identify and evaluate the overlap (or discrimination) between two samples from the same population. Also, it is useful for evaluating data from samples with only a reduced number of observations. The method has been used to analyze both specific-interspecific and specific variations and has been employed in different disciplines, such as Palaeontology [132], Archaeology [133], and Forensic Anthropology [134]. In this case, this statistical analysis allowed us to evaluate whether the differences between the averages of the facet angles obtained from our sample is significant enough to distinguish typological categorizations (specifically, the PSSB groupings). And to examine discrimination on inter-group and intra-group levels.

## Stage 4.) Analysis of the type and position of percussion traces

In previous papers, we have highlighted the intensive percussive activities carried out by hominins occupying the BL site [51, 52, 135]. Our work on these limestone percussion tools, complemented by experimental analysis, has allowed us to catalogue the variability of traces of percussion in terms both of their type and their position, and with respect to the morphology of the cobbles. This methodology, outlined in Barsky et al [51], served to study damage patterns on the tools presented in this paper. Importantly, we observe whether percussive traces occurred before, or after, the different phases of reduction for each tool, thus establishing their temporal relationship with respect to the negatives and cortical surfaces on each piece.

## Results

Results from seven multipolar and multifaceted limestone tools from the BL site are presented using the methodology presented above (Tools A to G, Table 2), and highlighting: (1) the different degrees of surface alteration of the limestone; (2) the general morphological and dimensional characteristics of each tool; (3) the core reduction and volume management strategies used; (4) the distinction of groupings within the sample as determined from statistical variability of the amplitude of the facet angles; (5.) the identification and characterization of percussive activities carried out with the sample tools and their relationship to the different phases of management.

### 4.1 Degree of surface alteration

The sample of seven limestone multifaceted tools with multipolar removal negatives (Tools A to G, Table 2) presents visibly varying degrees of silicification, as well as different degrees of taphonamically-induced alteration. Tools A and B, have a high degree of surface alteration (Level 4. Fig 3D.), with phenomena of rounding and peeling that has deeply affected the original morphology of the interfaces between the negatives. Tool C presents a medium to high level of alteration (Level 3. Fig 3C.). In addition, its surface has been particularly modified by percussive activity that occurred after all of the different phases of management. Meanwhile, Tools D, E, F and G show only low or medium degrees of alteration (Level 1 or Level 2, Fig 3B.).

**Table 2. General morpho-technical characteristics of the analyzed multifacial multifaceted tool sample from BL.**

| Tool | Alteration | Size (L/W/T) | Weight | Volume cm³ | Faciality | Polarity | General morphology | Min. nº negatives | Average (L/W) of all negatives (dimensions and Standard Deviation) | | Num whole negatives | Average (L/W) of whole negative (dimensions and Standard Deviation) | | Facet angle average |
|---|---|---|---|---|---|---|---|---|---|---|---|---|---|---|
| A | 4 | 70 x 60 x 55 mm | 300 gr | 113,10 | Multifacial | * | Polyhedral-rounded | ** | *** | | *** | *** | | 124° |
| B | 4 | 69 x 68 x 60 mm | 411 gr | 153,13 | Multifacial | | Rounded | | | | | | | 133° |
| C | 3 | 70 x 63 x 54 mm | 298 gr | 113,10 | Multifacial | Multipolar | Polyhedral-rounded | 16 | 29 mm (SD: 10,97) | 29 mm (SD: 8,36) | *** | *** | | 128° |
| D | 1 | 92 x 84 x 72 mm | 743 gr | 280,34 | Multifacial | Multipolar | Rounded | 16 | 25 mm (SD: 8,81) | 31 mm (SD: 9,27) | 6 | 34 mm (SD: 11,10) | 30 mm (SD:10,2) | 137° |
| E | 2 | 86 x75 x 72 mm | 637 gr | 255,38 | Multifacial | Multipolar | Rounded | 17 | 34 mm (SD; 10,60) | 32 mm (SD: 10,90) | 10 | 33 mm (SD: 10,04) | 31 mm (SD: 9,51) | 132° |
| F | 1 | 82 x 80 x 70 mm | 665 gr | 250,44 | Multifacial | Multipolar | Cuboidal | 17 | 37 mm (SD: 10,04) | 31 mm (SD: 7,88) | 8 | 37 mm (SD: 6,32) | 29 mm (SD: 8,86) | 126° |
| G | 1 | 90 x 67 x 67 mm | 468 gr | 173,96 | Trifacial | Multipolar | Polyhedral | 16 | 25 mm (SD: 13,04) | 24 mm (SD: 12,20) | 9 | 17 mm (SD: 10,71) | 17 mm (SD: 8,47) | 126° |
| | | 80 x 71 x 64 mm | 502 gr | 191,35 | Average | | | 16.4 | | | | | | |

Tools listed A to E all present a polyhedron or subspheroid morphology, Tools F and G are polyhedral cores used for comparative purposes. In some cases, surface alteration has impeded the interpretation of: (*) the knapping directions; (**) the number of removals, and in some cases the recognition of whole negatives (***).

## 4.2 General dimensional features

Overall, the tools analyzed show dimensional variability, as determined by differences in their size, weight and volume (Table 2). All of them present at least 16 readable negatives (excluding minor negatives that may have resulted from separate phases of retouching). The average length and width of the negatives (both whole and transected or partial) varies between (L) 37 mm and (W) 24 mm; while whole negatives only present a length and width average of (L) 37 mm and (W) 17 mm (Standard Deviation = ~10) (Table 2 and "S1 Dataset").

The average of the angles forming the crests separating the negatives (calculated by trigonometry) on each piece indicates that some of the tools present wider (or more abrupt) angles separating the facets (> 130), compared to others (< 130) (Table 2 and "S2 Dataset").

For the pieces with visible impact points (N = number of angles with visible impact points for each piece), we calculated the average of the knapping angles (kn.a): Tools D and E (134°,

N kn.a = 7 and 131˚, N kn.a = 8, respectively) showed more obtuse angles than for Tools F and G (121˚, N kn.a = 9 and 116˚, N kn.a = 4, respectively).

It is important to underline that, while the final morphologies of the tools occur as cuboid, polyhedral, polyhedral-rounded or rounded, the analysis of the original shape of the cobbles, as assessed from the form of the surfaces displaying residual cortex, indicates *a priori* anthropic selection of rounded cobbles compared to the range of cobble forms available in the sedimentary context of the BL site [51, 52].

## 4.3 Management strategies: Results of the diacritical analysis

The diacritical analysis allowed us to reconstruct the different phases involved in the manufacture for four of the tools (Tools D, E, F, G) (Fig 7, Fig 8, Fig 9 and Fig 10). Unfortunately, a high degree of alteration impeded the diacritical reading of two of the tools (Tools A and B), rendering it impossible to recognize superimposition between their removal negatives (necessary to establish sequentiality). Finally, the poor surface condition of Tool C allowed only a partial analysis of this piece (Fig 11). The results of the diacritical reading of each tool are illustrated, described and compared below.

**Tool D (Fig 7):** Three phases of manufacture can be identified in the diacritical analysis: Phase 1 (indicated in green): involved the creation of a percussion platform (Profile A). Phase 2 (indicated in yellow): some negatives forming the platform were orthogonally truncated as the piece was knapped along its periphery (Profiles B and E). Because they are not contiguous, it is impossible to determine which of these two profiles was knapped first. Phase 3 (indicated in blue): a series of removals effectuated from a cortical platform (Profile D) irregularly cut into the structure transversely (Phase 2) and frontally (Phase 1), removing some negatives from the anterior phases. The use of the bipolar-on-anvil method is evidenced (in particular on Profile B) (see negative 5 * and 6 * in contrast to negative 7 *); with some negatives probably resulting from the repercussion of the anvil (for example impact point of negative 7*). Traces of percussion (Fr. *piquetage*) observed on the cortical surface (Profile C—circle) could reflect active percussion activity. Alternatively, they may be related to passive percussion; resulting from impacts as the piece was secured on the anvil to facilitate its reduction.

**Tool E (Fig 8):** The diacritical interpretation of this tool reveals three distinct phases of management: Phase 1 (indicated in green): the platform (Profile B, green) was created by a removal. It is unclear whether it was made before or after the peripheral removals, indicated in geen in Profile E and F. In Phase 2 (indicated in-yellow) extractions labelled with numbers 4 *, 5a, 5b, 5c, 6, 7, 8 (Profiles A, C, E and F) cut into Phase 1 negatives, effacing evidence of overlaps and impact points. The tool was peripherally knapped using the percussion platform created by negative 1* (Profile B). Interestingly, Phase 3 (indicated in blue), involved removals that create the rounded morphology of the tool (Profiles C and D). In this case, it is impossible to determine whether the tool was knapped using the bipolar-on-anvil method, or by free-hand technology.

**Tool F (Fig 9):** Four distinct phases of management are highlighted by the diacritical analysis. Phase 1 (indicated in green): Creation of the main knapping platform (Profile A) and a few removals on the periphery (Profile D). As in Phase 2a (indicated in light yellow), a Siret type negative (negative 5, profile A and D) with impact along its periphery breaks obliquely into the overall morphology of the tool, creating a contact area separating the two perpendicular surfaces defined in Phase 1. Phase 2b (indicated in dark yellow) clearly corresponds to a new phase of peripheral knapping, wherein a series of parallel removals orthogonally cuts the surfaces defined in Phases 1 and 2a. It is impossible to determine temporal order between the Profiles (B and E) since the negatives, although continuous and overlapping, are not contiguous

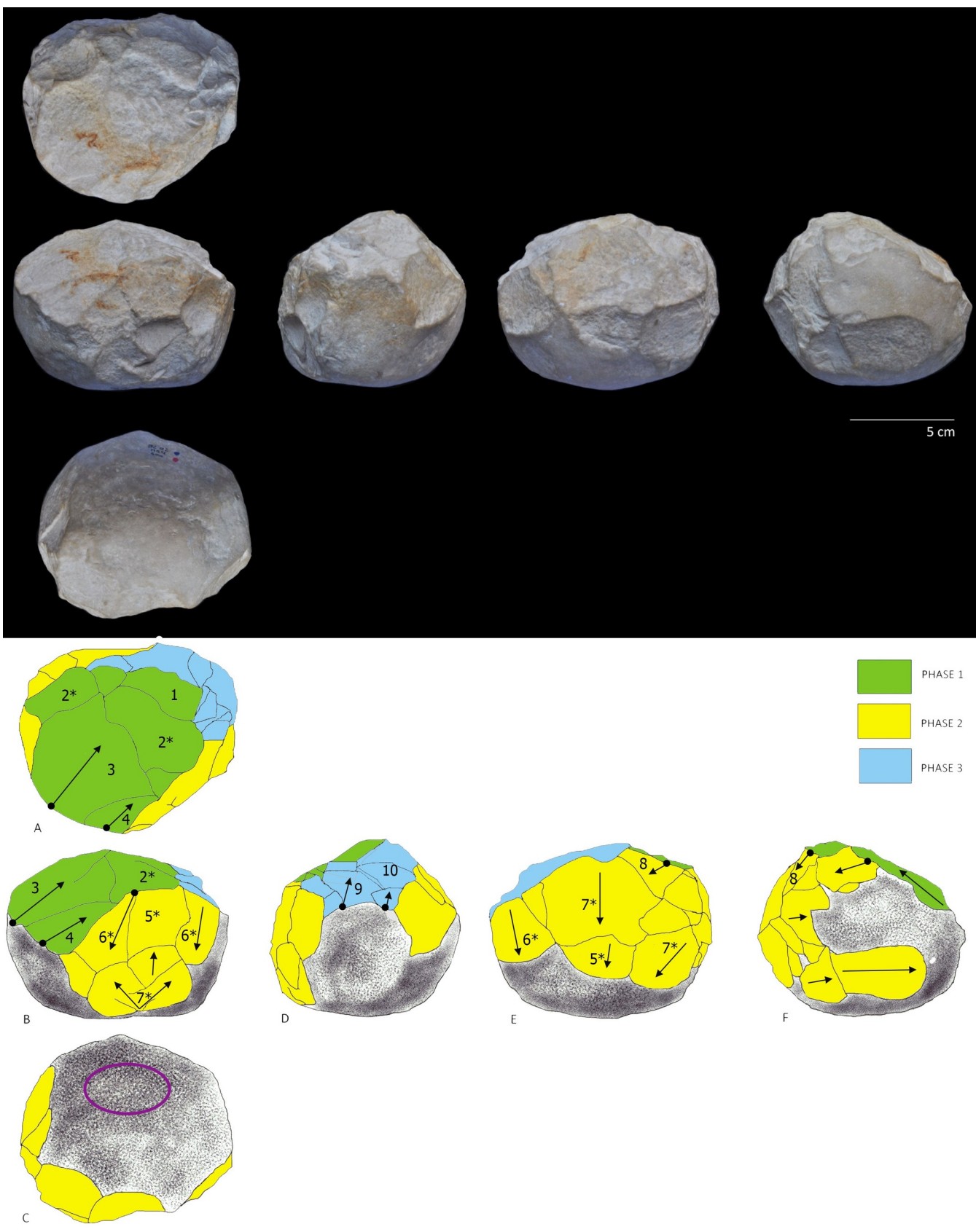

PHASE 1

PHASE 2

PHASE 3

**Fig 7. Photo and diacritical drawing of Tool D.** Phase 1 (green) creation of a percussion platform (Profile A). Phase 2, (yellow) Platform negatives transected orthogonally, as the piece was peripherally knapped (Profiles B and E). Phase 3 (blue): Series of removals from a cortical platform (Profile D), cutting the structure transversely (Phase 2) and frontally (Phase 1). The use of the bipolar-on-anvil technique is observed (in particular on Profile B) (see negative 5 * and 6 * in contrast to negative 7 *), with some negatives probably resulting from anvil repercussion (ie. negative 7*). Traces of percussion (Fr. *piquetage*) are observed on the cortical surface (Profile C—circle).

between the two surfaces (separated by Phase 1 and Phase 2a as in profile D). Phase 3 (indicated in light blue) involves a change of knapping direction, with a series of removals cutting orthogonally into the negatives of Phase 2b (Profile C). Finally, Phase 4 (indicated in dark blue) constitutes retouching from the Phase 1 platform that overlaps the Phase 2b negatives. It is unclear whether this final phase of retouching was effectuated before the Phase 3 extractions since there is no contact between the affected surfaces. Clear use of bipolar-on-anvil flaking is visible on Profiles B and E.

**Tool G (Fig 10):** In spite of the absence of impact points of the removal negatives forming Profile A (indicated in green) and Profile B (indicated in yellow), the diacritical analysis of this tool allows to appreciate the anteriority of the latter as an initial phase of platform preparation (Phase 1) for the former (Phase 2); clearly reflective of an orthogonal management system. Negatives with cortical percussion platform (Phase 3, indicated in light blue) cut the preceding phases in different directions. A final phase (percussion marks indicated in purple) is materialized by accidental removal negatives from convergent cortical surfaces, attributed to active percussion (Profile C and E). The use of this tool for percussion is posterior to the previous phases of management since these accidental negatives, attributed to a last phase, transect those of the previous Phase 2.

**Tool C (Fig 11):** This tool displays a semi-peripheral edge particularly affected by percussive activity that occurred following a phase of core management. The intensity of this percussive damage has obliterated the extraction departures, thus eliminating the possibility of obtaining their order with any confidence. The diacritical reading allows to clearly discern two phases of life for this tool, in spite of the impossibility of determining temporal order within the management phase: 1) a knapping phase and 2) a pounding phase (indicated in yellow and purple, respectively, Fig 11) where they have been recognized as *micro-removals* and *facetted breakage* [52].

Although the BL study sample is limited, similarity in the management strategy of multifacial morphologies with multidirectional negatives is identified. In fact, all of the tools analyzed show systematic exploitation. Comparison of the management systems extrapolated from the diacritical readings allows to observe that, for all of the pieces, there was an initial knapping phase oriented towards the generation of a main percussion platform (Phase 1). Subsequently, a second phase of knapping was effectuated from this platform, using the orthogonal knapping system (Phase 2). During this second phase, the original cobble morphology was peripherally cut, creating different surfaces (Profiles) represented by several contiguous negatives, that were, however, not necessarily sequential and overlapping. A third phase is defined by the rotation of the tool and a new phase of oblique knapping which appears (1) as a configuration phase since it lends the final sphericity to the tools (Tool D and E; Fig 7 and Fig 8) or (2) as a continuation of orthogonal management (Tool F an G; Fig 9 and Fig 10). This operative scheme is common to all of the pieces of our study sample (Fig 12).

In some cases (Tools E, F and G), Phase 1 platform negatives are observed on two profiles (Fig 12) and it is impossible to determine which of these two surfaces preceded the other since they are separated by posterior removal negatives. Tools F and G present more intensive stages of cobble reduction, with Phase 1 apparently corresponding to an earlier knapping stage; rather than simply creating a platform by way of one or two blows. In any case, this previous

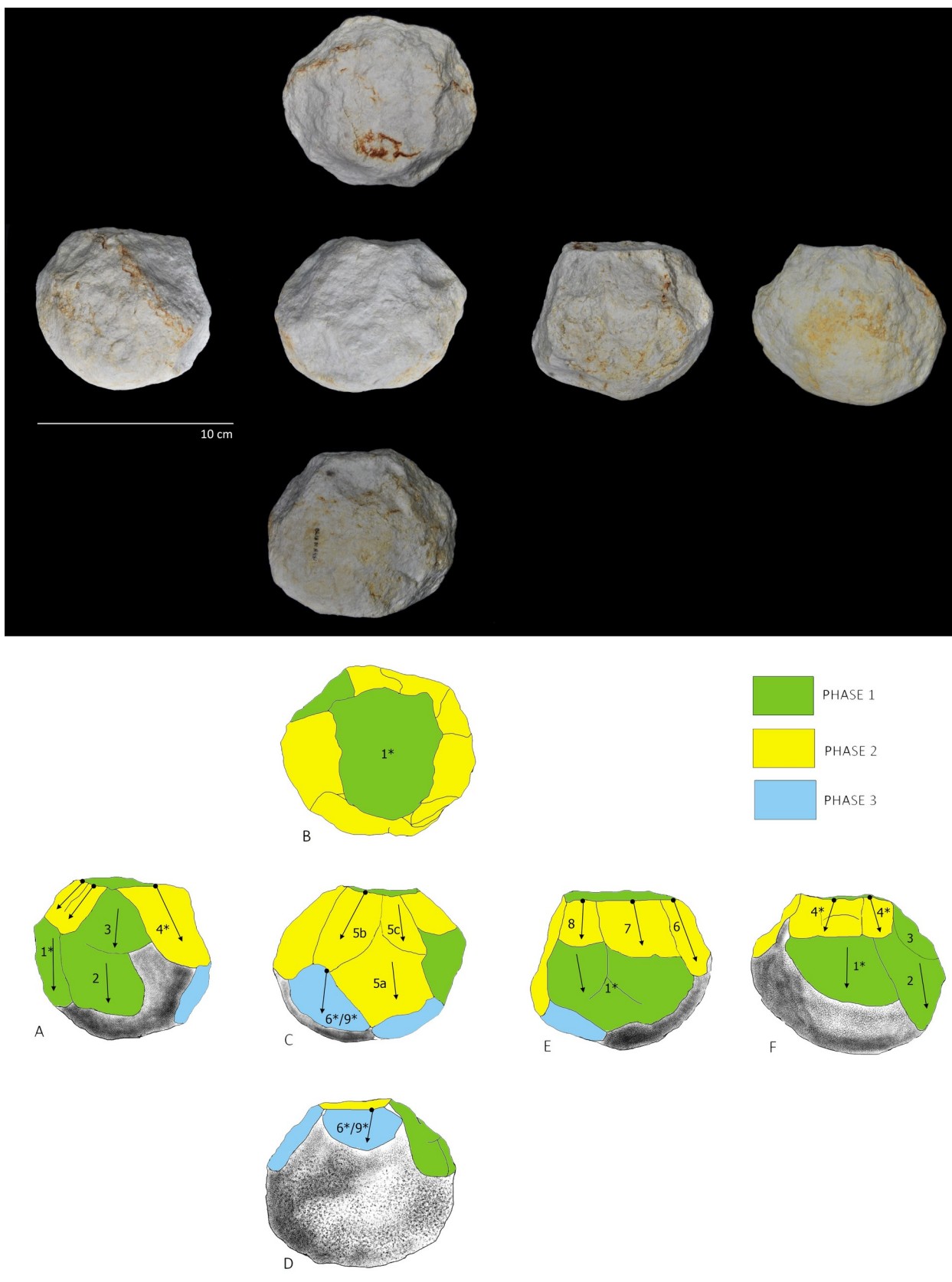

**Fig 8. Photo and diacritic drawing of Tool E.** Phase 1 (in green): Creation of a platform (Profile B) by a removal or a previous phase of partial peripheral management of the tool (Profile E and F, green); Phase 2: (yellow). A series of peripheral extractions 4 *, 5a, 5b, 5c, 6, 7, 8 (Profiles A, C, E and F) from percussion platform created by negative 1* (Profile B) partially effaces the Phase 1 negatives. Phase 3 (blue), oblique removals contribute to lending a rounded morphology to the tool (Profiles C and D).

knapping phase, evidenced by the diacritical analysis, is only partially represented in both cases.

The cortical residue preserved on Tools D and E allows us to recognize the original cobble morphology, indicating that only minor changes were made affecting its initial shape. Importantly, the flakes attributed to this final phase were so thin that they would either have broken during extraction, or been useless for any cutting activity: this implies that *the aim of these removals was not production*. Rather, it seems that the exploitation was intended to add facets following the rounded morphology of the initial cobble.

Other observations from this stage of our analysis include: 1) The use of the bipolar-on-anvil method was sometimes used to manage the cobble reduction peripherally, as is evident on Tool D (Profile B), and Tool F (Profile B) and; 2). Posterior phases of peripheral retouching (Tool F) with oriented average size of the retouch is 10 mm x 13 mm (respectively, with Standard Deviation 2,99 and 5,10); 3) In addition to the retouch, some accidental removal negatives resulting from the use of the tools for active percussion activity are recognized, indicating different life-phases for these tools.

## 4.4 Morphometric analysis

The observation of the results supported by the statistical analysis of each group of angles indicates two main tendencies that generate a similar pattern between two groups (Table 3), clearly observable in a boxplot diagram (Fig 13). In both groups, the Standard Deviation is less than half the angles *Mean* (Tab. 3), which allows us to suppose low variability. However, variability is recognized in the width of the internal angles of each piece. In fact, two main tendencies are observed, concerning: rank, mean, Standard Deviance (SD), variance and median:

- **Group A: Tools A, C, F and G** present a range greater than 40, between more than 100–150 degrees, with an average $< 128$ degrees, a standard deviation $> 10$ and a variance above 130 (Tools A and C present a variance of more than 170);

- **Group B: Tools B, D and E**, present a range $< 33$, between 119 and 153 degrees, with an average of $> 131$ degrees, a standard deviation $< 10$ and a variance less than 85 in all cases, present a wide angle of quite homogeneous and relatively obtuse angles.

Groups A and B are distinguished by internal variability of the width of the angles, determining the presence of tools with smaller angles and different amplitudes (Group A), and tools with more open angles and with less dimensional variation in terms of the amplitude of the angles (Group B). This characteristic of the second group gives the objects a more rounded shape than those of the first group, which, contrastingly, tends to present more angular and irregular profiles.

The comparison of the angles of the two groups using the Lubishew's test allows to identify internal similarities between the analyzed morphologies. Among the groups we obtain a value R = 0.382, and a degree of discrimination 64.9%. Although the Lubishew's test indicates only a moderate degree of discrimination between groups, descriptive statistics highlights a morphological distinction between the forms, linked to the width of the angles.

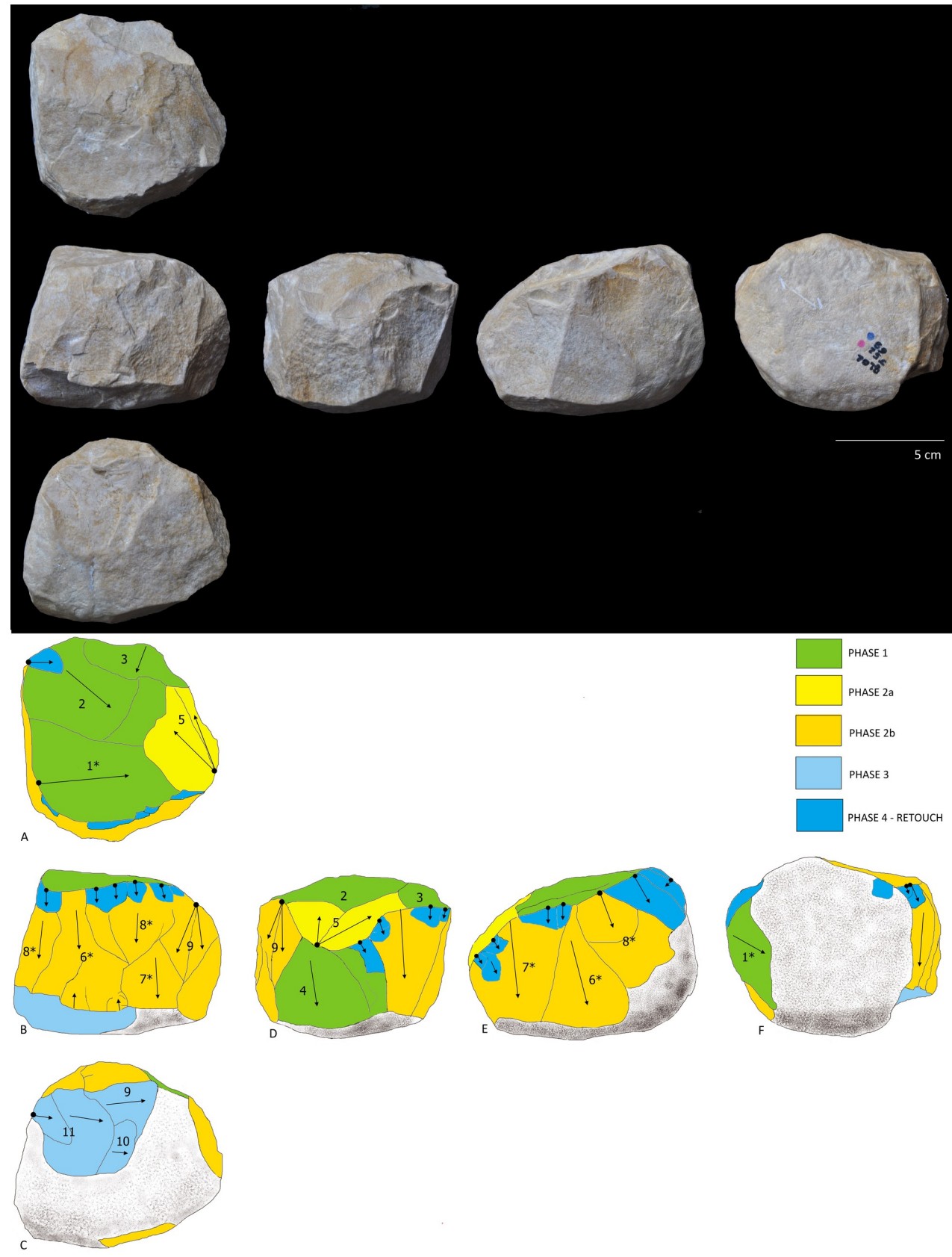

**Fig 9. Photo and diacritical drawing of Tool F.** Phase 1 (green): Peripheral removals (Profiles D) and creation of the main knapping platform (Profile A). Phase 2a (light yellow): a Siret flake (negative 5, Profiles A and D) was obtained obliquely, creating a contact area separating the two surfaces defined in Phase 1. Phase 2b (dark yellow): a new series of parallel, peripheral removals was made, departing from the platforms defined in Phases 1 and 2a (Profiles B and E) (between them Phase 1 and Phase 2a). Phase 3 (light blue): the cobble was rotated to knap a few flakes that cut into the surface of the Phase 2b negatives. Phase 4 (dark blue): retouching from the Phase 1 platform overlaps the Phase 2b negatives.

## 4.5 Traces of percussion

Four of the tools present traces of percussion (terminology from [51, 52]: *Micro-removals* (Fig 14A) and *facetted breakage* (Fig 14E) were recorded on different trihedral interfaces between negatives. *Accidental removal negatives* are located on a natural trihedral surface (Fig 14D).

*Surface scarring* (Fig 14B) and *cupola* damage (Fig 14C) are observed, respectively, on flat non-cortical surfaces (platform generated by a removal) and cortical semi-rounded cobble surfaces.

Traces of percussion occur on different surface morphologies: trihedral (non-cortical and cortical); flat or rounded cobble surfaces (non-cortical and cortical); likely indicating different kinds of uses for these tools, within a range of potential percussive activities [52].

Flat cobble surfaces with traces such as cupula, or other kinds of surface scarring, can be linked to anvil knapping activities. On the tools presented here, these traces are localized respectively on platform A (upper surface) (positioning criteria [51, 136, 137]) and platform and B (lower surface), and are not observed on the cobble extremities.

Traces of percussion identified on the tips of the trihedral zones, such as facetted breakage, may be associated with active hard percussion activity (see experimental work in [52]); micro-removals could be the result of repeated pounding of soft materials on an anvil, producing retouch and crush marks on abrupt edges, as shown in the case of the BL heavy-duty scrapers (see experimental in [135]). Multi-functionality and re-use are also characteristics of the BL Oldowan assemblage [66, 138], where hominins appear to have opportunistically selected cobbles with rounded morphologies, perhaps because they were found to be suitable for holding in a fist. They recall ellipsoidal hammers and present abrupt edges shown to be useful for secondary pounding activities [52, 135]. Because these marks were found to have been made posterior to the knapping phases, this 'secondary' percussive activity places these tools within a different link in the operative schemes; relating to use rather than manufacture.

## 5. Discussion

Advances have been made in defining what is meant by the term 'Oldowan' [18], and new analyses of assemblages attributed to this techno-complex [31, 139, 140] have contributed to reopening the debates on the associated cultural phase defined by Mary Leakey [13] as 'Developed Oldowan' [8]. These debates concern the validity of maintaining internal subdivisions within the Oldowan techno-complex, and whether or not the DO could be considered a transitional phase towards the Acheulian (as recognized by Leakey in DOB).

The elimination of the DOA in favor of the term late Oldowan [33] has been proposed, as significant technological innovations were not clearly perceived in assemblages with this denomination. With respect to Classic Oldowan, some authors observe an increase in percussive activities and preferential raw material selection [30]. In addition, other analyses include materials initially attributed by Leakey [13] to the DOB within the Early Acheulian, giving importance to the handaxe frequency [8, 31, 32, 141].

While some sites, HWK EE and EF-HR, initially considered DOB by Leakey [13], have been reinterpreted in accordance to these criterion (mainly variability in pounding tools and large flake production) as Oldowan and Acheulian, respectively [139, 140]. Currently, in sites

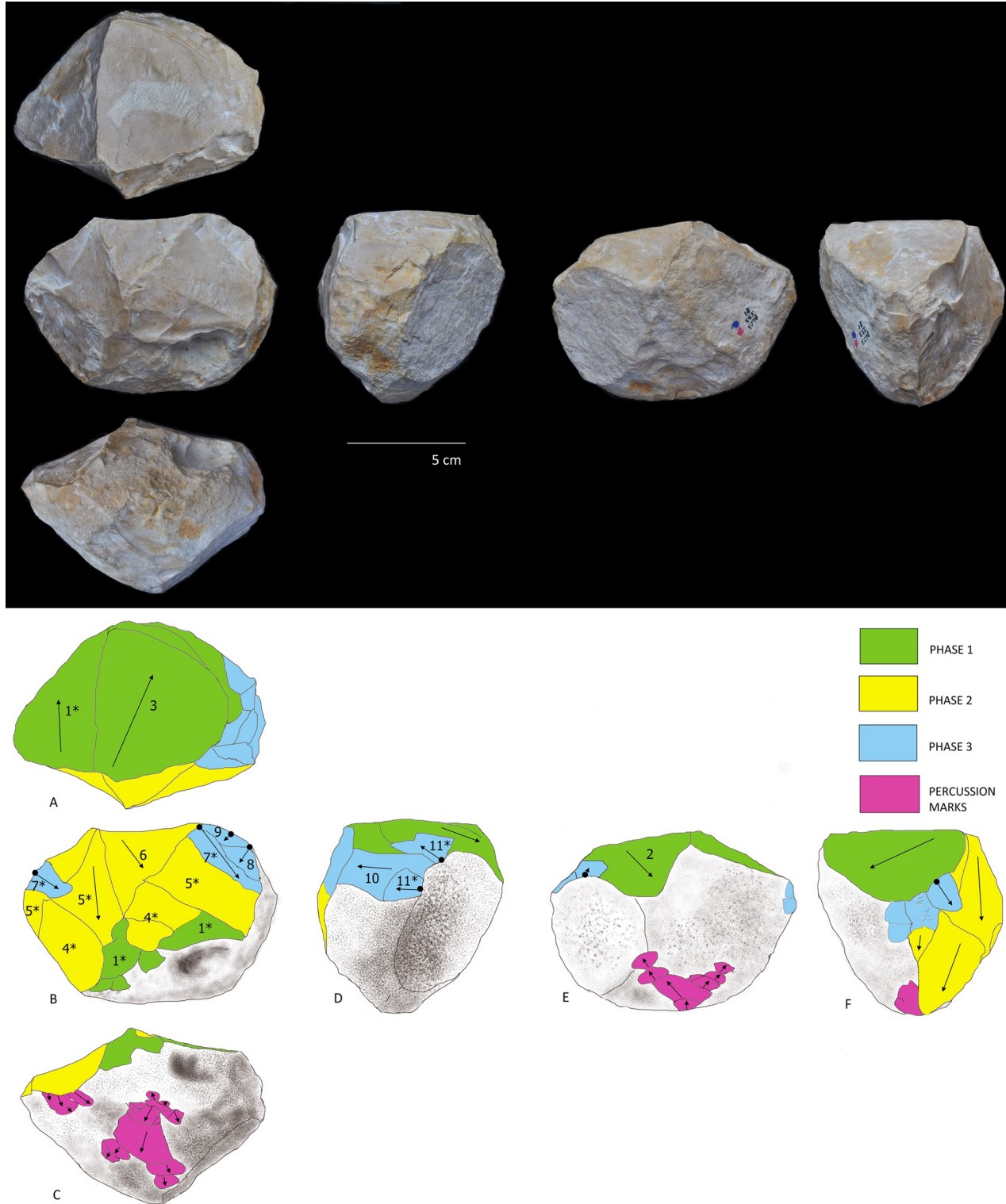

**Fig 10. Photo and diacritical drawing of Tool G.** Phase 1 (green): creation of the percussion platform (Profile A). Phase 2 (yellow): recurrent knapping series of flakes using orthogonal method from Phase 1 platform. Phase 3 (light blue): Tool rotation and small removals from cortical surface cutting previous negatives of Phases 1 and 2. A last phase (use) corresponds to accidental removals (purple), on convergent cortical surfaces and on Phase 2, due to percussion activity.

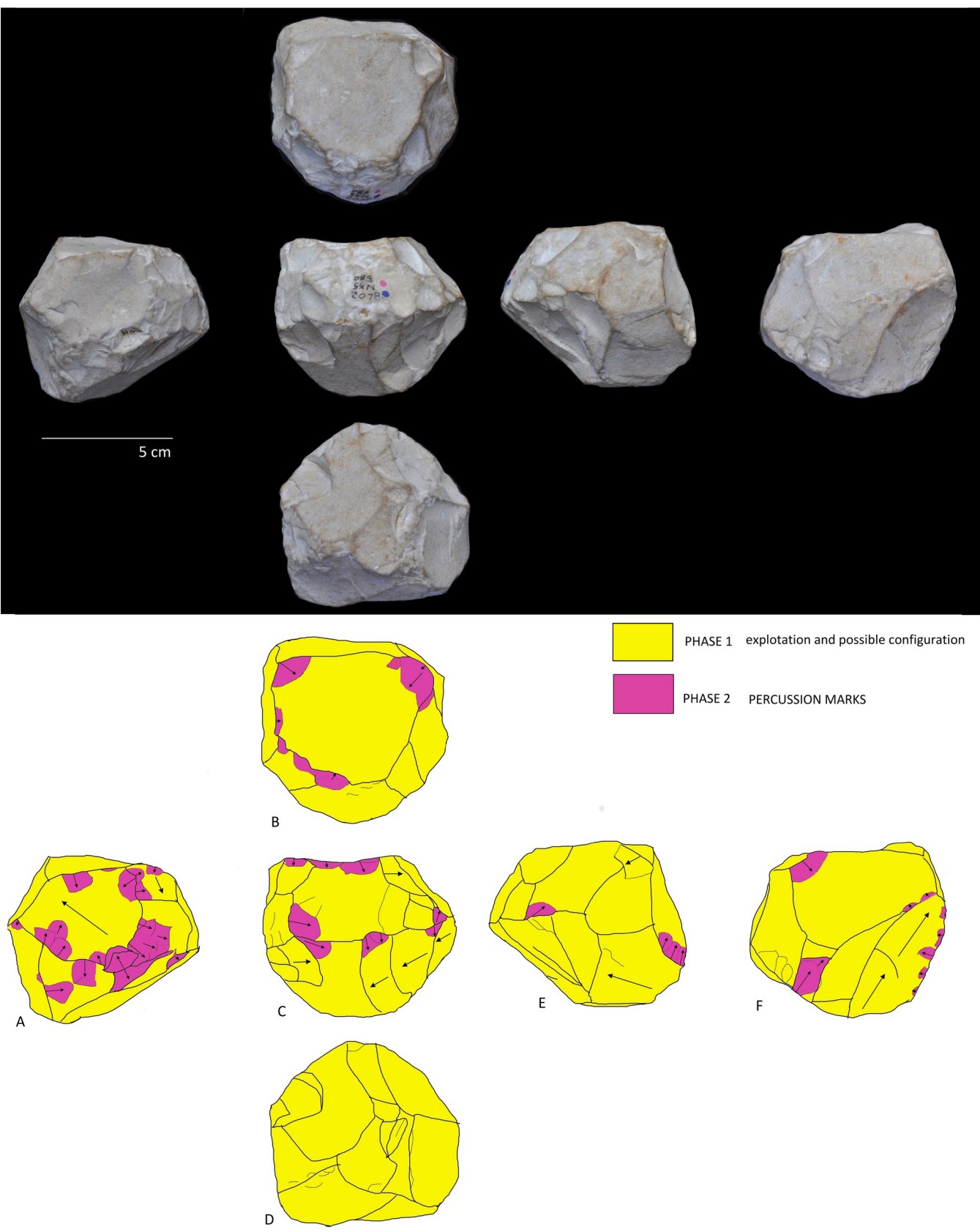

**Fig 11. Photo and diacritical drawing of Tool C.** Phase 1 (indicated in yellow): A single phase of core management (or possibly configuration). Phase 2 (indicated in purple): the tool's protruding crests were used for pounding activity.

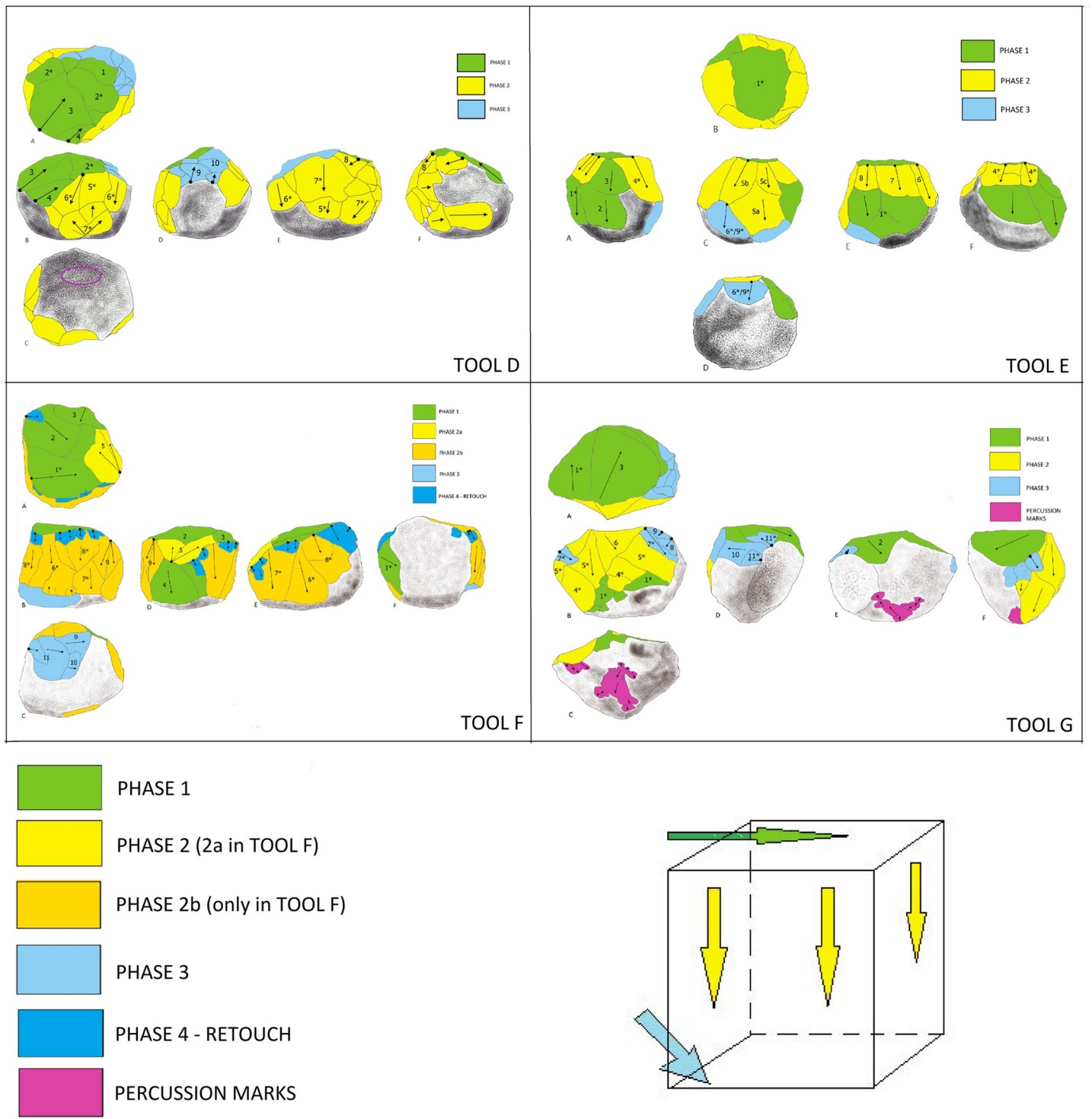

**Fig 12. Comparison of the diacritical drawings and simplified operative scheme deduced from the diacritical lecture (cube on bottom right).**

**Table 3. Univariate statistics of the different tools from BL site.**

| Tool | N | Range | Min | Max | X | se | SD | cv |
|------|-----|-------|--------|--------|--------|------|-------|-------|
| A | 12 | 42.28 | 103.21 | 145.49 | 124.45 | 4.11 | 14.22 | 11.42 |
| B | 10 | 22.40 | 124.10 | 146.50 | 133.26 | 2.65 | 8.38 | 6.29 |
| C | 20 | 44.93 | 104.08 | 149.01 | 127.57 | 2.94 | 13.13 | 10.29 |
| D | 24 | 32.24 | 119.75 | 151.99 | 137.04 | 1.87 | 9.18 | 6.70 |
| E | 16 | 30.96 | 121.81 | 152.77 | 131.76 | 2.18 | 8.72 | 6.62 |
| F | 19 | 40.65 | 106.63 | 147.28 | 125.59 | 2.76 | 12.03 | 9.58 |
| G | 15 | 44.24 | 101.68 | 145.92 | 126.16 | 2.95 | 11.43 | 9.06 |

N: number; Range; Min: minimum; Max: maximum; X: mean; se: standard error of the mean; SD: Standard Deviation, cv: coefficient of variation. Coefficient of variation (cv) is calculated as the ratio of the standard deviation (sd) to the mean (X) and is a measurement of dispersion in relation with the mean of a sample. Therefore it is a standardized measurement of dispersion. All the cv are relatively low which it could be interpreted as a certain standardization. Nevertheless, we can establish two different groups. Those tools with a cv between 9 and 11 (Group A: Tools A, C, F, G) and those close to 6.00 (Group B: Tools B, D, E). The highest value corresponds to that of A and the lowest one to B.

with a low frequency of proto-bifacial tools, the absence of large flakes, handaxes and cleavers, the idea that the DOB represents a transitional phase to the Acheulian is no longer accepted. Therefore, the redefinition of DOA and DOB (DOC is unclear and generally included in DOB; [8]) as, respectively, Classic Oldowan and Early Acheulian, has resulted in the term Developed Oldowan falling into disuse and the abandonment of the division between DOA and DOB [142]. Spheroids, when interpreted as simple hammers or used cores, lose their significance as cultural markers. However, if we consider them as tools resulting from an intentional manufacture process, then their significance should be maintained. When the PSSB are accompanied by Acheulian elements in an assemblage, their cultural attribution is moved from Oldowan to Acheulian (Table 4).

In Africa, sites with PSSB morphologies are generally situated within a chronological range between 1.8 and 1 Ma. Elsewhere, they are documented in more recent contexts (Table 4), where they may be associated or not with handaxes and framed accordingly into either the Acheulian or the Oldowan techno-complexes. Contrastingly, in sites without handaxes attributed to the Oldowan, PSSB are associated with percussive tools. Beyond two tools with multi-faceted morphologies (flint polyhedrons), documented by Bisi and colleagues [104] found on the surface at Ca' Belvedere di Monte Poggiolo (Emilia Romagna, Italy, *ca*. 1 Ma.; [105, 107], the only European Oldowan site with subspheroid morphologies is BL.

Oldowan assemblages are generally described to contain tools of relatively simple elaboration, but reflecting a considerable level of technical and cognitive complexity [27, 176, 177], allowing it to persist through space and time. The Oldowan, framed within a long chronological period, has been described by some as a time of cultural stasis. Its duration on a global level is of 2 Myrs [21], combining the African data, with a duration of 1 Myrs (2.6–1.75 Ma.) [141, 178, 179]; with Asia where it was maintained from 1.2–0.9 Ma; and Europe, where it is recognized from 1.4–1 Ma. The oldest African industries do not have spheroid morphologies: Lomekwi 3 (3.3 Ma., West Turkana, Kenya; [180]), Bokol Dora 1, (BD 1) (2.61 and 2.58 Ma., Ledi Geraru, Ethiopia; [10]), Kada Gona and Ounda Gona (2.6 Ma., Afar, Ethiopia, [9, 24], AL 666 (2.3 Ma., Hadar, Ethiopia; [181, 182, 183, 184]; Fejej FJ 1a (Ethiopia 1.96 Ma.) [29, 185, 186] (see list, African Oldowan localities in [1]. Still others have introduced terms such as Pre-Oldowan [116, 187], to designate assemblages older than the Oldowan from Olduvai Gorge, without spheroids and lacking standardized retouched tools. This is not to be confused with

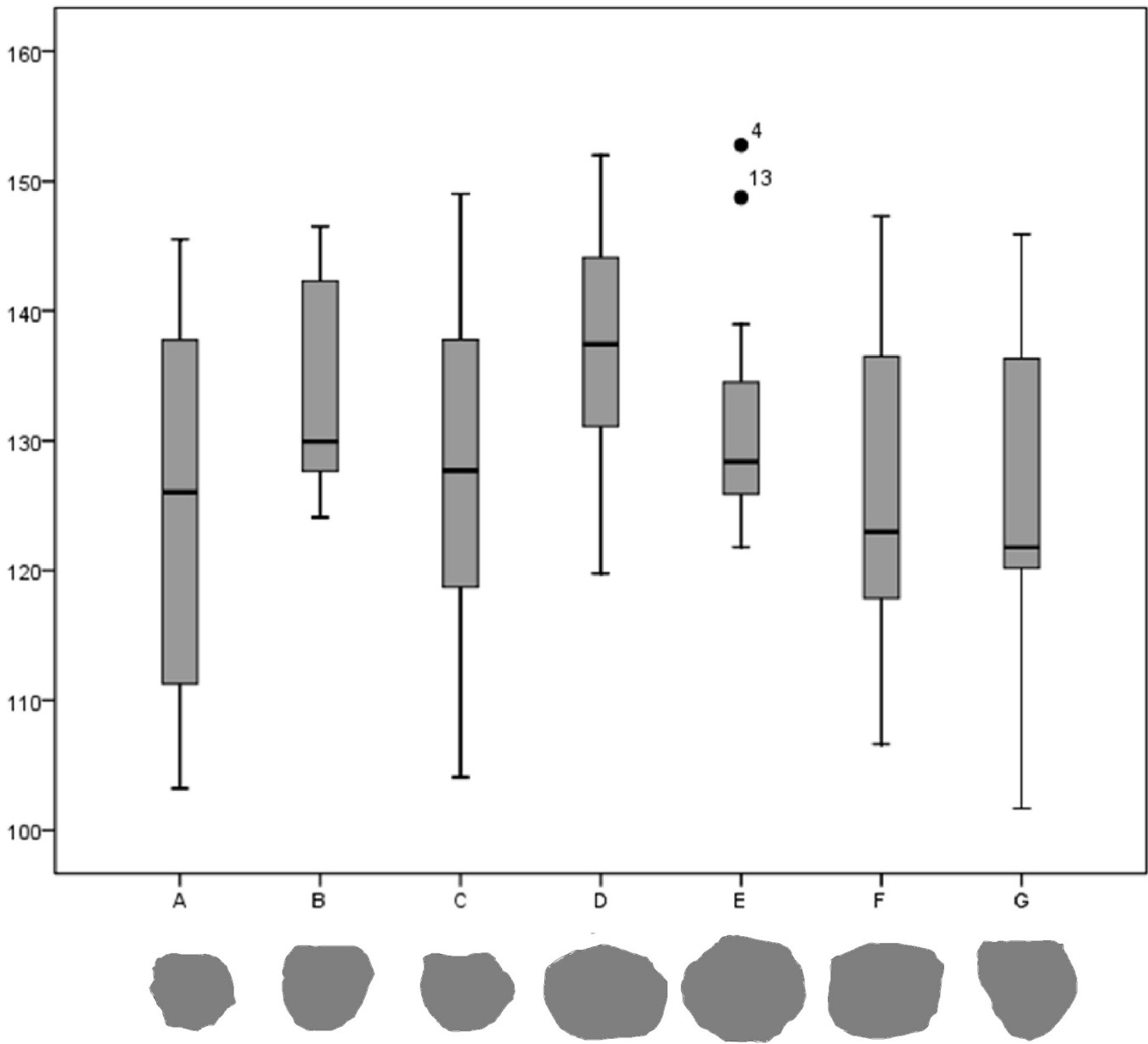

**Fig 13. Boxplot underlining the dual tendency of the pieces.** Group A (Tools A, C, F and G) compared to Group B (Tools B, D and E).

Mode 0, which refers only to a hypothetical phase of tool use preceding recognizable, systematic manufacture [28].

Of course, this discrepancy in the nomenclature reflects variability within the Oldowan techno-complex. We propose that the appearance of recurrent morphotypes such as the PSSB indicates change with respect to previous systems. So, even if the DO terminology is eliminated and replaced by the term late Oldowan, we observe that there are changes in the lithic toolkits after around 1.8 Ma.; in close chronological range with the first Acheulian [1].

Compared to the older assemblages, the ability of hominins to pursue a recurrent strategy in the management of multifacial multipolar cores, as in the BL subspheroids described here, represents a step forward in terms of cognitive advancement. The presence of recurrent

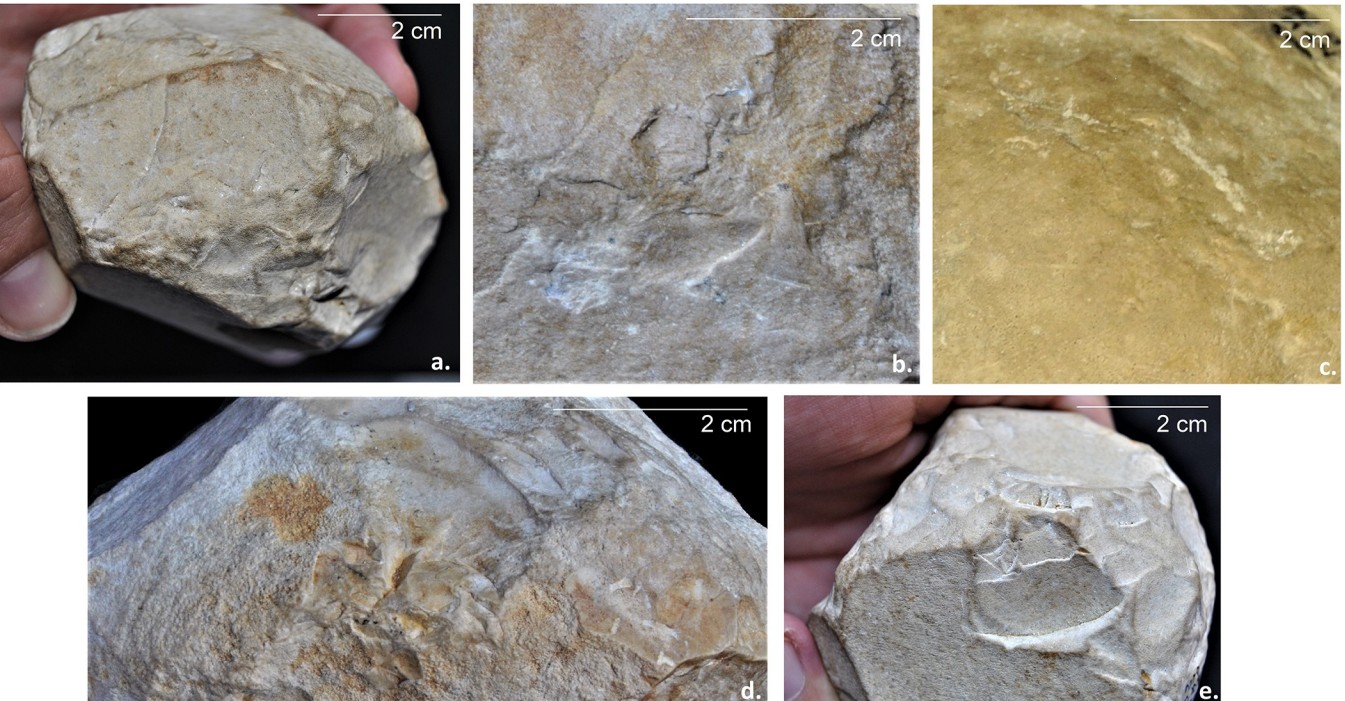

**Fig 14. Percussion marks on the analyzed tool sample.** a.) Micro-removals; b.) Surface scarring; c.) Cupola; d.) Accidental removal negatives; e.) Facetted breakage.

morphotypes in late Oldowan assemblages (PSSB, HDS, [135]), underlines different levels of complexity [188], moving beyond the more restrictive interpretations of the Oldowan (example: [189]).

It therefore appears essential that we better understand the technologies used to make spherioid morphologies, in order to distinguish between associated forms present in the assemblages, such as: (I) hammerstones with active edges or with fracture angles, which are documented as products of recurrent percussion activities affecting overall tool morphology by creating facets and percussion marks [136, 141]; *versus* (II) spheroids or subspheroids that present multiple knapping surfaces [31, 139, 141].

Although these morphologies may finally present formal characteristics lending them a spherical shape, with a volumetric structure organized around a central point, they are products of distinct activities, justifying their attribution to a separate tool category from any intentionally knapped (shaped) morphotype.

The presence of knapping surfaces observed on spheroids and subspheroids has been defined as representative of multifacial, multipolar core types with rounded morphologies [31, 37, 41, 42, 125]. Contrastingly, intentional spheroid manufacture with a true operative scheme has also been proposed [34]. Our study shows that the latter may be a viable proposition in cases where a predetermined configuration or a repeated reduction technique can be recognized. Such cases are, however, dependent upon the possibility of effectuating an accurate diacritical reading that may be carried out when surface structures have not been significantly altered, either by taphonomic phenomena or by subsequent active percussion activities; and this is not always the case.

The distinction between the polyhedral, spheroid and subspheroid categories has been made, in some cases, through the identification of an independent operating scheme, as

**Table 4. Sites with PSSB morphologies divided by geographic zone and Oldowan or Early Acheulian techno-complexes.** Oldowan or Early Acheulian techno-complexes with PSSB morphologies [13, 16, 31, 34, 36, 37, 38, 41, 42, 46, 51, 104, 105, 107, 118, 119, 120, 125, 136, 139, 141, 143, 144, 145, 146, 147, 148, 149, 150, 151, 152, 153, 154, 155, 156, 157, 158, 159, 160, 161, 162, 163, 164, 165, 166, 167, 168, 169, 170, 171, 172, 173, 174, 175].

| Thecnocomplex atributed by bibliography | Zone | | | |
| --- | --- | --- | --- | --- |
| | Africa | Levantine corridor | Asia | Europe |
| Acheulian | **Sterkfontein** STK-M5E (South Africa, 2.18–1.7 Ma.) [143]; **Swartkrans** SWT-M1 (South Africa, 2.19–1.8 Ma.) [144, 145]; **Melka Kunture a Garba IV**, Gomboré, Karre (Awash valley, Ethiopia 1.5 Ma.) [16, 146]; **Olduvai Bed II**, in **BK** [139, 147]; **TK; MNK Main; SHK; EF HR sites**; (Tanzania, between 1.2 Ma. and 1.353 ± 0.035 Ma.) [31, 139, 148, 149, 150]; **Gadeb 2 and Gadeb 8** (Ethiopia, 1.45–0.7 Ma.) [141], before DO, [151, 152]; **Chesowanja** (Kenya, 1.42±0.07 Ma.) [141, 153], before DO in [154]; Oldovan in [155]; **Olorgesailie** (Kenya, 1 Ma.) [156, 157]; **Isenya** (Kenya) [34, 157, 158, 159]; **Erg Tihodaïne** (Algeria, "…*Middle Pleistocene age*") [160]; **Isimila** (Tanzania, dated 260 Kya) [161, 162] | 'Ubeidiya (Israel, 1.6 Ma.) [120]; **Qesem Cave** (Israel, 420–200 ka.) [36, 163]; **Gesher Benot Ya'aqov** (Israel, 0.78 Ma.) [164] | **Jijiawan, Xiazhaijiacun** and in **open air sites of Lantian region** (close to Gongwangling site) (Lantian, central China, 600–300 ka.) [165, 166, 167]; **Dingcun** (North China, 160–210 ka.) [168]; **Dondrapal site** (Bastar region, Central India; 50 ka,—~7000 yrs. ago) [169]; **Ngebung** (Sangiran, Central Java, Indonesia; 0.8 Ma.) [170, 171, 172]; Central Syria **Hummal** (El Kowm, Central Syria; 1.6–1.2 Ma.) [173, 174] | **Bois-de-Riquet site- Unit 4** (Lezignan-la-Cebe, Herault, France, 1.2 Ma) [119]; **Roussillon** Quaternary **terraces** (Occitania, France) [46]; **Unit 1 of Santa Ana cave** (Extremadura, Spain, 117–183 Ka. ca., BP) [118] |
| Oldowan | **Ain Hanech** (Algeria, 1.8 Ma.) [37, 41, 42, 125, 175]; **Ounjougou** (Bandiagara, Dogon Country, Mali) [38]; **Olduvai Bed I** in **FLK North**, levels 1–6; **FLK "Zinjanthropus"** Floor; **FLK NN**, levels 1-2-3; **DK** levels 1,2,3 (Tanzania) [13, 31]; **Olduvai Bed II: FC West** [136]; **MNK skull site; FLK N** Sandy Conglomerate, **HWK East level 3-4-5, HWK EE (OGAP)** [139]; HWK EE [13] | | | **BL** (Spain, 1.4 Ma.) [51]; **Ca' Belvedere di Monte Poggiolo** (Emilia Romagna, Italy, *ca.* 1 Ma.) [104, 105, 107] |

indicated by De Weyer [38] for the site of Ounjougou (Mali). In some sites, these categories may even be represented by different raw materials, underlining differences in *chaîne opera-toires* [136]. For example, at Olduvai Gorge, polyhedron morphologies tend to be manufactured from lava, while pieces defined as spheroids are in quartz. Meanwhile, at Isenya, polyhedron and subspheroid morphologies were knapped in phonolite, and bolas are made from quartz [190]. This distinction related to raw materials is not observed at BL, where the PSSB morphologies identified so far are all made from limestone cobbles.

In this study, the differentiation between polyhedrons, subspheroids and spheroids is not, therefore, based on criteria relating to raw materials, but rather has been made using the width of the angles separating the facets of each piece. We define polyhedron morphologies by higher variability in their angles (Fig 13), which are also more acute, likening them to the group of multifacial cores. Comparatively, truly subspheroid morphologies display more open facet angles and greater uniformity.

The polyhedral and subspheroid morphologies in the BL lithic assemblage do not result from continuous battering. While they have been found to differ from the hammerstones with fracture angles, also present in the assemblage [52], they reflect an organized operative scheme that is somewhat comparable to the management of orthogonal cores. On the BL tools, traces

of active percussion, when detected, are posterior to the knapping phases. Interestingly, their dispersion and damage concentration patterns do not seem to be related to knapping activity, but rather to the treatment of a medium to low resistance material. For example, semi-peripheral micro removals (Figs 11 and 14) are comparable to those obtained experimentally by pounding dried tendons on a stone anvil [135]. Also, trihedral angles such as those on polyhedrons are not efficient hammers for stone knapping (see experimental in: [52]). The use of spheroid cores for the treatment of medium to hard materials has also been recently attested at HWK EE [139].

The diacritic analysis has allowed us to evaluate that, at least for the BL site, sub-spheroid morphologies result from a well-reasoned, organized shaping process that sometimes involved the use of a stone anvil (Fig 7). Meanwhile, such *preconceived morphological templates* transposed onto stone are usually attributed to the Acheulian, concretized, for Isaac [191], by the presence of innovative handaxes and standardized tools with recurring morphologies [192]. This justifies some DOB sites being re-assigned from the Oldowan to the Acheulian by de la Torre [141]. The concept of acquisition and transmission reflected in any systematic production of forms is qualified as culture. Even though the creation of a platform and subsequent peripheral management of a surface may seem simplistic, as in the case of orthogonal knapping, it became emblematic of Oldowan culture [135]. The subspheroid morphologies and the multidirectional cores of BL, for their repetition in the first phases of the management, can't be fit in the classic opportunistic flake production. Furthermore, we have to take in account that, for the site under consideration, not only the two subspheroids (Tools D and E) show evidence of shaping, because other intentionally repeated morphotypes are present: as for the case of heavy-duty scrapers which have been defined also by intentional shaping [135]. This feature becomes more characteristic from the Acheulian techno-complex.

In the case of the BL, we differentiate between the hammerstones with fracture angles and subspheroid morphologies that we have shown result from a pre-conceived desire to create a spherical or subspherical shape. As such, perhaps these tools with spherical symmetry could indicate a concept inherent to the late Oldowan, indicative of cognitive changes, just as tools with a bifacial and bilateral symmetry herald the Acheulian. The recognition of spheroids as intentionally manufactured tools can only be achieved by considering each site separately, depending on the possibility -or not- of performing a diacritical analysis that can highlight different phases of manufacture and even, perhaps, a preconceived formal structure.

## 6. Conclusions

Spheroids and subspheroids in ancient stone toolkits remain today morphologies that are not fully understood; both in terms of their functionality, and from a technological point of view: whether they result from an organized or random management or a predetermined concept. With the exception of two flint polyhedral pieces reported from Ca' Belvedere de Monte Poggiolo [104], this study documents, for the first time, subspheroid morphologies in the European Oldowan context. Also, the present study offers a new, multidisciplinary analytical methodology, which combines information from raw materials, technological feature analysis and geometric morphometric analysis, contributing to the study and comparison of spherical morphotypes present in ancient stone toolkits, highlighting the evaluation of similarities and differences between the different accepted categories defined by the PSSB typological grouping.

So far as the polyhedral and subspheroid morphologies from BL are concerned, it appears clear from our analysis, that these tools are the result of a systematic management of rounded cobbles. While the organization of the removal negatives reflects the same type of management

as that observed on the multidirectional cores, they differ in that the last phase of reduction, which was not effectuated orthogonally in relation to the previous knapping phases, but rather by a different gesture, aimed at creating more open facet angles, following the morphology of the initial rounded cobbles and giving the final subspheroid form.

Unlike the relatively uniform management system recognized for the subspheroids from BL presented here, PSSB morphologies recognized in other Oldowan and Acheulian sites may, in the future, prove to result from diverse modes of management. In any case, their diacritical readings, will vary depending on post-depositional alterations or traces of percussion that alter the readability of the intersections separating the facets. However, the diacritic analysis presented here has proven to be a fundamental method for extrapolating the management methods used in the shaping/knapping of PPSB, that will be an important step forward in shedding light on different cultural attitudes that led to the generation of these tool forms. While measurement of the facet angles provided another important criterion by which to distinguish analogous morphotypes forming the different categories of the PSSB group. In this case, for example, a difference was noted between cores and the polyehedrons with subspheroid morphologies, with respect to variability and the obliquity of the angles which, when more open, gives the tools a more rounded shape.

In the future, it will be interesting to analyze larger samples of PSSB morphologies using the methodology presented here, in order to shed light on hypotheses about whether such forms derive from the interest of obtaining a certain volume around a central point. Dispersion of the mass with respect to this point could indicate differences between the categories, while corticality remains a distinguishing factor separating subspheroids and spheroids. Only the study of a larger sample and the comparison between different collections using this methodology can help to discern the different components forming the PSSB grouping.

## Supporting information

**S1 Dataset.**
(XLSX)

**S2 Dataset.**
(XLSX)

## Acknowledgments

This research was funded by the Junta de Andalucía, Consejeria de Educacion, Cultura y Deporte: Orce Research Project "*Primeras ocupaciones humanas y contexto paleoecológico a partir de los depósitos pliopleistocenos de la cuenca Guadix-Baza*: *zona arqueológica de la Cuenca de Orce (Granada, España), 2017–2020*"; "*Presencia humana y contexto paleoecológico en la cuenca continental de Guadix-Baza. Estudio e interpretación a partir de los depósitos Plio-Pleistocénicos de Orce. Granada. España*" B120489SV18BC, 2012–16; "*Primeras ocupaciones humanas del Pleistoceno inferior de la cuenca de Guadix-Baza (Granada, España)*" B090678SVI8BC, 2009–11; MICINN (no feder) "*Estudio de las dispersiones faunísticas y humanas durante el Pleistoceno inferior en la cuenca mediterránea.*", CGL2016-80975-P, 2017–19; the Spanish government Ministerio de Ciencia, Innovación y Universidades (MICINN-FEDER) code CGL2016-80975-P, and the Generalitat de Catalunya Research Group 2017SGR 859. "*Comportamiento ecosocial de los homínidos de la Sierra de Atapuerca durante el Cuaternario V*", MICINN-FEDER PGC2018-093925-B-C32 and the Generalitat de Catalunya, AGAUR agency, SGR 859 and SGR 1040. We extend gratitude to the Gerda Henkel Foundation (AZ 32/V/19, Lower Paleolithic Spheroids Project (LPSP). ST is beneficiary of the

Provincia Autonoma di Bolzano (Italy) post-master scholarship. AB has been funded from the European Union's Horizon 2020 research and innovation programme under the Marie Sklo-dowska-Curie Action grant agreement PREKARN nº702584. The research of DB, JMV, & RSR is funded by CERCA Programme/Generalitat de Catalunya. JMJA belongs to the Research Group HUM-607. We are grateful to Leore Grosman and Gonen Sharon for their comments and the fruitful discussions about spheroid morphologies and to Fanette Lara Chmiel for the drone photography of the Barranco León site. We thank the Junta de Andalucía and the staff of the Archaeological and Ethnological Museum of Granada who facilitated access to the Barranco León lithic collection. We also acknowledge the researchers and students involved in the excavations, recovery, preparation, and study of the archeo-paleontological record from the Barranco León site.

## Author Contributions

**Conceptualization:** Deborah Barsky.

**Data curation:** Stefania Titton, Deborah Barsky, Amèlia Bargalló.

**Formal analysis:** Stefania Titton, Alexia Serrano-Ramos.

**Funding acquisition:** Robert Sala-Ramos.

**Investigation:** Stefania Titton, Deborah Barsky, Josep Maria Vergès, José García Solano.

**Methodology:** Stefania Titton.

**Project administration:** Isidro Toro-Moyano, Robert Sala-Ramos, Juan Manuel Jimenez Arenas.

**Resources:** Isidro Toro-Moyano.

**Software:** Alexia Serrano-Ramos.

**Supervision:** Deborah Barsky, Amèlia Bargalló.

**Visualization:** Stefania Titton.

**Writing – original draft:** Stefania Titton.

**Writing – review & editing:** Deborah Barsky, Amèlia Bargalló, Juan Manuel Jimenez Arenas.

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
