## [Decision Letter · Decision Letter 0]

4 Oct 2019

PONE-D-19-22634

Subspheroids in the lithic assemblage of Barranco León (Spain): recognizing the late Oldowan in Europe

PLOS ONE

Dear Mrs. Titton,

Thank you for submitting your manuscript to PLOS ONE. After careful consideration, we feel that it has merit but does not fully meet PLOS ONE’s publication criteria as it currently stands. Therefore, we invite you to submit a revised version of the manuscript that addresses the points raised during the review process.

Though both reviewers felt that this paper had merit, both agree that the paper is thin on description and documentation in several key parts of the paper. The reviewers also point out a number of specific areas of improvement. It is also noted that the paper should be thoroughly read by a native English writer if it is to be resubmitted. I agree with their overall assessment and I hope that the paper is significantly bolstered and revised. 

We would appreciate receiving your revised manuscript by Nov 18 2019 11:59PM. To enhance the reproducibility of your results, we recommend that if applicable you deposit your laboratory protocols in protocols.io, where a protocol can be assigned its own identifier (DOI) such that it can be cited independently in the future. For instructions see: http://journals.plos.org/plosone/s/submission-guidelines#loc-laboratory-protocols

We look forward to receiving your revised manuscript.

Kind regards,

Michael D. Petraglia, Ph.D.

Academic Editor

PLOS ONE

Journal Requirements:

2.  In your manuscript, please ensure you have provided sufficient information regarding the specimens used in your study to allow others to replicate the analyses. Ensure that you have reported specimen numbers and complete repository information, including museum name and geographic location.

For more information on PLOS ONE's requirements for paleontology and archaeology research, see https://journals.plos.org/plosone/s/submission-guidelines#loc-paleontology-and-archaeology-research.

4. We note that [Figure(s) in your submission contain [map/satellite] images which may be copyrighted. All PLOS content is published under the Creative Commons Attribution License (CC BY 4.0), which means that the manuscript, images, and Supporting Information files will be freely available online, and any third party is permitted to access, download, copy, distribute, and use these materials in any way, even commercially, with proper attribution. For these reasons, we cannot publish previously copyrighted maps or satellite images created using proprietary data, such as Google software (Google Maps, Street View, and Earth). For more information, see our copyright guidelines: http://journals.plos.org/plosone/s/licenses-and-copyright.

1.    You may seek permission from the original copyright holder of Figure(s) to publish the content specifically under the CC BY 4.0 license.

Additional Editor Comments (if provided):

Reviewers' comments:

Reviewer's Responses to Questions

**Comments to the Author**

1. Is the manuscript technically sound, and do the data support the conclusions?

Reviewer #1: Yes

Reviewer #2: Partly

2. Has the statistical analysis been performed appropriately and rigorously? 

Reviewer #1: Yes

Reviewer #2: Yes

3. Have the authors made all data underlying the findings in their manuscript fully available?

Reviewer #1: Yes

Reviewer #2: Yes

4. Is the manuscript presented in an intelligible fashion and written in standard English?

Reviewer #1: Yes

Reviewer #2: Yes

5. Review Comments to the Author

Reviewer #1: This study presents some important findings on spheroids artifacts: it shows that these subspheroids were the product of a systematic management of rounded cobbles (line 829); polyhedron morphologies van be defined by higher variability in their angles, which are also more acute (783) and these morphologies in the BL lithic assemblage do not result from continuous battering (786).

The conclusion states: In (the) future (846), it will be interesting to analyze larger samples of PSSB morphologies using the methodology presented here, in order to shed light on (the) hypothesis about whether such forms derive from the interest of obtaining a certain volume around a central point. Dispersion of the mass with respect to this point could indicate differences between the categories.

However, why this hypothesis should be of importance is not fully discussed. The most effective solid projectile has a spherical form in which the most mass can be concentrated into the least dimension, so this hypothesis is assumed to include the probable use of these tools as projectiles.

The mass of the tools is given in Table 2 is consistent with examples from the natural spheroids of the Cave of Hearths and which have been shown to be good throwing material in terms of impact energy and muscle power (Wilson et al 2016, Cannell 2002, Cannell 2018), with an average of 502 g – similar to the Cave of Hearths value of 533 g. Although the number of samples does not allow for a mass distribution analysis, it would be of interest to examine their mass, together with that of the 51 percussion tools (active and passive) and 249 pebbles and cobbles without anthropic traces. Indeed, given the close proximity of the Fuente Nueva site and similarity of the limestone cobbles (whole and broken) and cores, (which make up about 75% of the 446 macro objects- tools and possible manuports) of the combined assemblage (ref. 48), the mass distribution of the subspheroids in relation to the two assemblages could add greatly to our comprehension of these tools.

This is of particular importance given the evidence for confrontation between hominins and large carnivores at Fuente Nueva, where an incomplete skeleton of Mammuthus meridionalis was found surrounded by 34 coprolites, 17 lithic artifacts, and 32 unmodified cobbles. The skewed spatial distribution of these elements, the physical characteristics of the coprolites, and the absence of the elephant limbs and cranium suggest that both hominins and hyenas scavenged the carcass of this megaherbivore, following a sequence of consumption in which the hominins arrived first, dismembered and transported the limbs, and probably also the cranium, and later the hyenas consumed the rest of the elephant carcass (Espigares et al., 2013).

It would also be of interest to note that similar Olduwan spheroids have been found at Ain Hanech, Algeria (Early Pleistocene and dated at about 1.8 Ma) and Hummal, Syria (in deposits dated to at least 1 Ma). In the former case the classification of many rocks as spheroids has been disputed; originally described as “sphéroides à facettes” or faceted spheroids (Balout, 1955), a subsequent analysis suggested that “the bulk of the assemblage was comprised of simple cores,” (Sahnouni, Schick, & Toth, 1997, p. 703). The present work and results of the close examination of the angles used may help to clarify the cataloguing of this largest assemblage of spheroid material.

At Hummal, two distinct mass distributions were found, one: «les gros, dont le diamètre est d’environ 8 cm pour un poids moyen de 540 g; les petits, d’un diamètre autour de 5,5 cm pour un poids moyen de 150 g.” The spheroids of Barranco are all in the former group, but there may be some samples that also fit this lighter mass, adding to the evidence of a single species selection with specific age or sex distributions.

To give more context to the site, it would help to give the full coordinates and presumed height above the lake surface. Level D1 is said to materialize a high-energy, rapid water transport of gravels and cobbles (198) so presumably this flood level was still well above the lake surface and the site protected at the rear by the rock face (?), but with open approaches that would need defense. This is of importance as it would allow another level of comparison with other sites where spheroid-like artefacts have been found.

Reference 109 is given as the source for examples at Ain Boucherit, Algeria, but it is surely worth mentioning the dating of the site of large polyhedrons at about 2.4 Ma, only 750 km distant from Barranco Léon, but separated by an immense time difference.

Some specific points (highlighted in red):

The naming of these tools should be constant and in the same case:

Fig 2. Polyhedral and subspheroid morphotypes from the BL site (a-e), and 280 multidirectional cores selected for the diachritical analysis (f-g).

455 Tools listed A to F all present a polyhedron or subspheroidal morphology, Tools G and H are polyhedral cores

From Fig. 2 and the diacritical analysis it appears that the cores are actually tools a and b (A and B) (?)

545 cortical surfaces, attributed to active percussion (Profile C and E). The use of this tool for (546) percussion in (is) posterior to the previous phases of management

561 temporal order within the management phase: 1) a knapping phase and 2) a pounding (562) phase (indicated in yellow and purple, respectively, Fig 11.). The figure uses the word ‘percussion’ rather than pounding, which implies hand-held striking. The marks on the rock’s angles could certainly be the result of several thrown impacts (training) and are unlike the (what appears to be) pounding marks in Fig. 10. The text states: (260) the limestone was used for different purposes; all mainly linked to percussive activities (261). A thrown rock is also percussive in that impact marks will show if it has been thrown and hit or landed on a hard surface.

721 Bisi and colleagues [100] found on the surface at Ca 'Belvedere de Monte Poggiolo (Ca’ Belvedere)

799 Sometimes the use of stone anvil attested (Fig 7)…. (Text missing ??)

802 innovation [188]. This justifies some DOB sites being reaassigned Acheulian de la Torre 803 [137]. (to the as stated by?)

The clipped document has the colour highlights

Reviewer #2: The paper presents a new addition on the lithic assemblage from a well-known European Mode 1 site, Barranco Leon. Specifically, it focuses on a small subspheroid assemblage made on limestone that has been analyzed through a four-phase technological perspective.

I was excited for reviewing this paper, but after finished I’m left with the feeling that more could have been done. Although the general presentation is correct, I've some inconsistencies and a remarkable scarcity on some of the paper sections, especially on the one related to the Morphometric analysis, which to me should have been the biggest asset of the paper. Therefore I recommend this paper to go through major revisions.

I also strongly recommend a general English revision of the manuscript. Maybe the authors can ask an English native colleague to review it for them. In addition, here is a list of some of the writing mistakes I was able to find during my revision:

Lin.156: “we was apply”

Lin. 268: “to remedy to the difficulties”

Lin. 176: “edorrheic”

Lin. 773: “be may”

Lin. 447: “visibles”

Lin. 613: “anges”

Lin.799: “Sometimes the use of stone anvil attested”

Lin. 835: “recongnized”

INTRODUCTION

Lin. 67: “… has been tentatively attributed to Kenyapithecus…”

I’ve never seen the word Kenyapithecus published anywhere. Kenyanthropus platyopts is the name given to the species in its original paper and it’s also how appear in the several publications that mention it. To mention a couple:

Leakey, M. G., Spoor, F., Brown, F. H., Gathogo, P. N., Kiarie, C., Leakey, L. N., & McDougall, I. (2001). New hominin genus from eastern Africa shows diverse middle Pliocene lineages. Nature, 410(6827), 433.

Spoor, F., Leakey, M. G., & O'Higgins, P. (2016). Middle Pliocene hominin diversity: Australopithecus deyiremeda and Kenyanthropus platyops. Philosophical Transactions of the Royal Society B: Biological Sciences, 371(1698), 20150231.

Lin 78-79: “… generating a process of distinction from other primates, underlined by the development of technical skills.”

It’s probably due to the way this sentence is written, but it looks that the authors are suggesting the mentioned process of distinction is somehow a conscious one. Furthermore, enough evidences of technology in chimpanzee contexts have been provided, so I would try to be careful with this kind of statements.

Lin. 87-91: “… Through time, the transition towards new reduction systems occurred within the OIC [28], progressively moving beyond the mechanics of the unifacial and unidirectional strategies, and leading to greater morpho-technological variability within some of the Oldowan tool kits, with orthogonal, unifacial discoid and multidirectional core management strategies…”

This statement is something to take into consideration. Is the suggested transition a real transition? Advanced Oldowan reduction strategies can be found in OGS-7, Lokalalei 2C and Kanjera South, to mention some, and all those sites predate 2.0 Ma. The Oldowan manifests a large regional variability in its complexity, both synchronic and diachronic, so I find risky, to say the least, the suggestion of a lineal progressive evolution. It doesn’t seem to be supported by the current state of the art. I know the authors have quoted this paper later on the text, but I find it essential in order to work through this paragraph.

Stout, D., Semaw, S., Rogers, M. J., & Cauche, D. (2010). Technological variation in the earliest Oldowan from Gona, Afar, Ethiopia. Journal of Human Evolution, 58(6), 474-491.

Lin. 118 – 121: “Based on these descriptions, the different categories comprising the PSSB can be distinguished from each other by direct observation in order to recognize deviances from the regular morphological-volumetric aspect of a sphere, developed around a central point (center of mass), which confers them a more or less rounded shape.”

To me, that is just a way to recognize the inherent subjectivity of any non-quantitative lithic analysis.

MATERIALS AND METHODS

Lin. 258: “Cores are scarce and intensively reduced (3.1 % of the 1 562 pieces in flint)”

How did the authors measure the reduction intensity of the cores? Which are the knapping strategies used? Do they correlate with the ones identified on the limestone assemblage? I feel like there is very few explanation regarding this topic, while a lot of information (maybe not that necessary) is provided on the general characteristics of the assemblage. I also get that is not exactly the point of this paper to present the core features, but they are the elements that could possibly be correlated with the following PSSB discussion.

Lin. 265: “After reviewing the entire lithic collection, five limestone pieces…”

The PSSB pieces sample is really small, five in total, a 0.2% of the whole lithic collection. I’m fully aware that there is nothing the authors can do about it; the sample is what it is, but they should be aware of the problem this sample poses if they intend to include the BL subspheroid assemblage into the large Oldowan discussion.

Lin. 268-271: “To remedy to the difficulties relating to surface preservation of these five pieces and to recognize and verify their special features and assess the direction of their removal negatives, we decided to compare them with 2 of the best preserved multifacial multipolar cores (out of the 10 available in the collection)”

I don’t fully understand why the authors did this. Establishing a comparison with multifacial multipolar cores implies that they thought about a relation between the two types of artifacts before the actual analysis of the PSSBs. If that is the case, I don’t understand why they limited the comparison to only two cores of the total assemblage, if taking all of them would have helped to enlarge the sample and obtain more reliable data. These also relates with my previous comment on why the authors didn’t provide with a larger explanation on the core’s features.

RESULTS

Lin.447-450: “For the pieces with visibles impact points (N=number of angle with impact point visible for each piece), we calculated the average of the knapping angles (kn.a): Tools D and E (134 °, N kn.a = 7 and 131 °, N kn.a = 8, respectively) showed more acute angles than for Tools F and G (121 °, N kn.a = 9 and 116 °, N kn.a = 4, respectively).”

How can 134 and 131 degrees angles been more acute than 121 and 116 degrees angles?

Lin. 451-453: “While the final morphologies of the tools occur as cuboid, polyhedral, polyhedral-rounded or rounded, the analysis of residual cortex indicates a priori anthropic selection of rounded cobbles”

These statement necessarily implies that all the selected pieces are volumetrically reduced at the exact same stage, which they don’t know or at least haven’t explain anywhere on the paper.

Lin. 459-460: “The diachritical analysis allowed us to reconstruct each phase of manufacture for four of the tools (Tools D, E, F, G) (Fig 7, Fig 8, Fig 9, and Fig 10)”

It might only be a matter of semantics, but I find that saying “each phase” might be a little bit risky. As they didn’t evaluate the reduction stage of the pieces, and assuming that new flake removals tend to erase the previous ones, I think it would be more accurate to say that they recognized the last phases of the sequence, not each one of them.

Lin. 467: “Tool D (Fig 7)…”

I don’t really see how the yellow surface presented in Profile E correlate with the green one presented in profile B, so its sequential ordination seems unclear to me.

Also, if a percussion platform is been created in Phase 1 (green), how the percussion platform for Phase 2 in Profile E appears to be the blue one (associated a Phase 3)? One possible explanation could be that the previous presence of Phase 1 removals were erased by the posterior Phase 3, but I cannot assure it just with pictures nor with the quality of the 3D scans. In any case, Figure 7 does not result very clear, and I recommend the authors to provide a more detailed explanation of the relationship between surfaces and Phases.

Lin. 492: “Tool E (Fig 8)…”

In Profile A, why aren’t the Phase 1 removals numbered? I don’t see any identification problem and I believe are the same ones marked on Profile F.

Lin. 511: “Tool F (Fig 9)…”

In general, not only on this tool, I think the graphic design on the diachritical schemes should be improved, as sometimes is leading to misinterpretations. For example, in Profile E, Phase 2b (dark yellow), the scar marked as 6* is clearly cutting into the one marked as 8*. Therefore, one would say that scar is posterior to 8* and not a previous one, when, judging by the actual photo and the authors explanation that is not the case.

Lin. 643-645: “This characteristic of the second group gives the objects a more rounded shape than those of the first group, which, contrastingly, tends to present more angular and irregular profiles”.

That is a little bit obvious, isn’t it? I really appreciate the intention of been thorough and pursue a quantitative approach with this paper, but the statement that a spherical volume has less acute angles than a polyhedron is hardly a result. In general terms, the morphometric analysis here presented looks a little bit scarce. These section of the paper is crucial to me and should be extended.

DISCUSSION

Lin. 709: “When the PSSB are accompanied by Acheulian elements in an assemblage, their cultural attribution is moved from Oldowan to Acheulian (Table 4)”.

Well, of course. It is a set of artifacts displaying diachronic presence. What is the point of remarking the artificial nature of any taxonomic category? Especially when those categories doesn’t take in account actual technological features, complexity parameters or quantitative data, but presence/absence of recognizable tools.

Lin. 718-719: “Contrastingly, in sites attributed to the Oldowan, PSSB are associated with percussive implements and there are no handaxes”.

This is, again, an example of circular reasoning that doesn’t say anything about the actual PSSBs.

Lin. 745-747: “The ability of hominins to pursue a recurrent strategy in the management of multifacial multipolar cores, as in the Orce subspherods described here, represents a step forward compared to the older assemblages in terms of cognitive advancement”.

There are older chronologies with long, systematized knapping sequences (See again Lokalalei 2C or even Gona). I don’t really see the step forward if we are referring to a set of lithic artifacts largely heterogenic that doesn’t present clear attributes of a final morphology. I find difficult to refer “PSSB” as a morphotype. It is a term that widely refer to diversity, but the elements that comprise them are quite different. So far, I’ve not seen any attempt to explain why that variability appear on the archaeological record.

Lin. 781-784: “We define polyhedron morphologies by higher variability in their angles (Fig 13), which are also more acute, likening them to the group of multifacial cores. Comparatively, truly subspheroid morphologies display more open facet angles and greater uniformity”.

Again, this statement relates with my previous comment on subspheroids being more rounded than polyhedrons on the Results section.

6. PLOS authors have the option to publish the peer review history of their article (what does this mean?). If published, this will include your full peer review and any attached files.

Reviewer #1: Yes: alan cannell

Reviewer #2: No

---

## [Author Response · Author response to Decision Letter 0]

6 Nov 2019

I upload as separate file: "Response to Reviewers" that responds to each comments.

Review: Subspheroids in the lithic assemblage of Barranco León (Spain): recognizing the late Oldowan in Europe

Response to Reviewer #1

This study presents some important findings on spheroids artifacts: it shows that these subspheroids were the product of a systematic management of rounded cobbles (line 829); polyhedron morphologies van be defined by higher variability in their angles, which are also more acute (783) and these morphologies in the BL lithic assemblage do not result from continuous battering (786).

We thank the reviewer for his appreciation and for underlining the importance of our research at the Barranco León site.

The conclusion states: In (the) future (846), it will be interesting to analyze larger samples of PSSB morphologies using the methodology presented here, in order to shed light on (the) hypothesis about whether such forms derive from the interest of obtaining a certain volume around a central point. Dispersion of the mass with respect to this point could indicate differences between the categories. However, why this hypothesis should be of importance is not fully discussed. 

One of the objectives of the methodology presented in our paper is that the study and presentation of the features of the subspheroid morphotypes from Barranco León be used for future work on larger samples from other sites. Our methodologies applied to PSSBs allow to evaluate the qualitative and quantitative variability of these morphologies (intra and inter-assemblage). Also, the diacritical analysis allows to distinguish, objectively, similarities and differences in core management systems of PSSBs that will be useful for comparisons between different Oldowan/Early Acheulian assemblages. The dispersion of mass around a central point is indeed one of special interest when analyzing manufacture processes and could be elaborated upon in another work dedicated especially to this theme. What our present work shows is that this process at Barranco León was intentional. We believe that the 5 pieces from Barranco León are significant since they served as a foundation for the elaboration and testing our combined methodologies. What is especially notable in our case study is the hypothesis of intentionality buttressed by our diacritical study, that shows a phase (phase 3) that does not seem linked to manufacture, but to shaping. In the Oldowan, such pre-conceived formal templates are extremely rare. The last twenty-five years of explorations on the Oldowan have revealed relative complexity in cognitive processes for knapping. However, very little is known about possible shaping processes. In this paper, we explore this potential cognitive step as a possible defining factor within the context of the so-called late Oldowan of Europe.

The theme of mass dispersion around a central point is an important aspect of spheroids in general, but our interest here is to move beyond subjective selective processes in defining what is and what is not, a spheroid morphotype, by applying an objective methodology. Also, the diacritical analysis serves to reconstruct operative schemes used at the Barranco León site, moving beyond manufacture to allow us to propose, in only 2 cases, a possible shaping process that is very significant in this chronology. 

The most effective solid projectile has a spherical form in which the most mass can be concentrated into the least dimension, so this hypothesis is assumed to include the probable use of these tools as projectiles.

This is of course a valid opinion and we do not contest it. However, as explained in the introductory section of our manuscript, there are numerous hypotheses about the functionality and manufacture processes of spheroids. We have therefore added this very important hypothesis in this section with the appropriate references: “The hypothesis that spheroid morphologies could represent projectiles has also been explored based on studies of mass distribution [48, 49, 50]”.

We have not, as yet, had the occasion to experiment with the throwing/projectile theory which, we argue (based on your publications), will be hard to adapt to the Barranco León site context because all of the cobbles at this site are part of the deposit itself. For this reason, one of our earlier projects was to elaborate ways to distinguish between anthopically used cobbles and natural cobbles (Barsky et al., 2015). Fist-sized rounded non-modified cobbles are abundant in the stratigraphy so that our question is: why the facets? If you have available abundant cobbles that you throw, why would you need to facet them? Our experimental work (Titton et al., 2018) also shows that natural rounded cobbles are well-suited to bone breakage tasks so that the facet question is not resolved by this issue either. Of course, non-modified fist-sized rounded cobbles could have been effective projectiles as well. In future, we hope to continue our research and explore and test the projectile hypothesis experimentally. Ideally on a site context where the cobbles are not integral to the depositional context. At Barranco León, the infill presents the full range of cobble forms and sizes: from very large flat slabs to smaller rounded cobbles.

The mass of the tools is given in Table 2 is consistent with examples from the natural spheroids of the Cave of Hearths and which have been shown to be good throwing material in terms of impact energy and muscle power (Wilson et al 2016, Cannell 2002, Cannell 2018), with an average of 502 g – similar to the Cave of Hearths value of 533 g. Although the number of samples does not allow for a mass distribution analysis, it would be of interest to examine their mass, together with that of the 51 percussion tools (active and passive) and 249 pebbles and cobbles without anthropic traces. Indeed, given the close proximity of the Fuente Nueva site and similarity of the limestone cobbles (whole and broken) and cores, (which make up about 75% of the 446 macro objects- tools and possible manuports) of the combined assemblage (ref. 48), the mass distribution of the subspheroids in relation to the two assemblages could add greatly to our comprehension of these tools. 

We appreciate the Reviewer’s interest in the topic of the projectile hypothesis. Firstly, we need to specify that, in the Fuente Nueva site, the limestone raw material context is slightly different, since there is mostly the use of block/block fragments; rather than cobbles. Average weight of the percussion material at Barranco León is of 640g (Titton et al., 2018); somewhat more than Cave of hearths (sic. reviewer’s published data). At Barranco León, some of the largest macro-tools on limestone cobbles give an average weight of 2224 gr.; underlining cobble variability at this site. While we would be interested to explore more this particular feature (weight), this is far from the main objectives of the present article: elaborate a methodology to study the PSSB category to differentiate each morphotype; determine intentionality by diacritical analysis, study angle crests as a criterion for understanding variability between the morphotypes, place the Barranco León site within a larger framework of the Oldowan. We would also like to stress that, although there is great spatial proximity between the sites of Fuente Nueva 3 and Barranco Leon, these sites present slightly different logistic situations that affected the choice of limestone raw materials. The Barranco León site is also somewhat older than FN3.

Of course, when we were treating different aspects of percussive activities at Barranco León, one of our first investigations was to seek out selective processes. We have already compared between cobbles with signs of clearly anthropic use versus non-anthropic in terms of morphometric features without evidencing significant results (Barsky et al., 2015).

In addition, because the infill corresponds to a flash food event, the calibration of the cobbles will vary in accordance to their place in the sequence due to gravitational and hydro-energetic factors. Another factor making this kind of analysis complicated at Barranco León is the variability of the conservation of the limestone: some of the cobbles are completely altered (as explained in the manuscript), while others display varying degrees of alteration: this greatly affects their weight. Also, different kinds of limestone are identified in the assemblage (Titton et al., 2018), with those more silicified being heavier than the marly. 

This is of particular importance given the evidence for confrontation between hominins and large carnivores at Fuente Nueva, where an incomplete skeleton of Mammuthus meridionalis was found surrounded by 34 coprolites, 17 lithic artifacts, and 32 unmodified cobbles. The skewed spatial distribution of these elements, the physical characteristics of the coprolites, and the absence of the elephant limbs and cranium suggest that both hominins and hyenas scavenged the carcass of this megaherbivore, following a sequence of consumption in which the hominins arrived first, dismembered and transported the limbs, and probably also the cranium, and later the hyenas consumed the rest of the elephant carcass (Espigares et al., 2013). 

The scenario explained here by the reviewer is from the Fuente Nueva site: not Barranco León. Of course, the faunal list between the two sites is analogous. However, we are presently working to demonstrate that the FN3 site represents a primary context, while the Barranco León accumulation is a secondary deposit that was posteriorly visited by hominins who then knapped both flint and limestone accumulated in this particular place: this will be published very shortly in collaboration with the geologists working on the Orce project. 

It would also be of interest to note that similar Olduwan spheroids have been found at Ain Hanech, Algeria (Early Pleistocene and dated at about 1.8 Ma) and Hummal, Syria (in deposits dated to at least 1 Ma). In the former case the classification of many rocks as spheroids has been disputed; originally described as “sphéroides à facettes” or faceted spheroids (Balout, 1955), a subsequent analysis suggested that “the bulk of the assemblage was comprised of simple cores,” (Sahnouni, Schick, & Toth, 1997, p. 703). The present work and results of the close examination of the angles used may help to clarify the cataloguing of this largest assemblage of spheroid material.

At Hummal, two distinct mass distributions were found, one: «les gros, dont le diamètre est d’environ 8 cm pour un poids moyen de 540 g; les petits, d’un diamètre autour de 5,5 cm pour un poids moyen de 150 g.” The spheroids of Barranco are all in the former group, but there may be some samples that also fit this lighter mass, adding to the evidence of a single species selection with specific age or sex distributions. 

While the average weight of the Barranco León pieces presented in this manuscript is close to that of Hummal and Cave of Hearths, these five pieces do show a range of weight from 743g to 298g. It is difficult, therefore, to argue, given the low number of pieces in the BL sample and this weight range difference, that there is a real similarity.

To give more context to the site, it would help to give the full coordinates and presumed height above the lake surface. Level D1 is said to materialize a high-energy, rapid water transport of gravels and cobbles (198) so presumably this flood level was still well above the lake surface and the site protected at the rear by the rock face (?), but with open approaches that would need defense. This is of importance as it would allow another level of comparison with other sites where spheroid-like artefacts have been found. 

Full coordinates of the site are: UTM 548400-4175340; 975 msnm. We do not see any use in publishing this in the manuscript because it has already been published elsewhere and does not add any significant information to the themes treated here. However, if the Editor sees fit, they can be included in the introductory section. During the occupation of the site, the flood level would have been almost level with the paleo lake. For more information on the geology of the Guadix-Baza basin, please refer to references: 53, 54, 56, and others. Whether or not the hominins living around the paleo lake threw stones because they were out in the open is difficult to ascertain. In addition, the extension of the BL and FN3 sites is presently unknown. We have, in previous pubications, already suggested that these sites represent points around the lake-marsh environment, where hominin activity has been evidenced archeologically: hopefully, in future, more such points will be discovered and studied. Throwing stones is not one of the activities that has been detected so far at the site, although we have experimented with the possbility that some of the larger cobbles might have been thrown to initiate debitage by the splitting or breaking them and thus opening up a platform.

Reference 109 is given as the source for examples at Ain Boucherit, Algeria, but it is surely worth mentioning the dating of the site of large polyhedrons at about 2.4 Ma, only 750 km distant from Barranco Léon, but separated by an immense time difference. 

With all due respect, the cited article does refer to the 2.4 Ma age: perhaps the reviewer would like to explain more?

Some specific points (highlighted in red):

The naming of these tools should be constant and in the same case: 

Fig 2. Polyhedral and subspheroid morphotypes from the BL site (a-e), and 280 multidirectional cores selected for the diachritical analysis (f-g).

455 Tools listed A to F all present a polyhedron or subspheroidal morphology, Tools G and H are polyhedral cores

From Fig. 2 and the diacritical analysis it appears that the cores are actually tools a and b (A and B) (?) 

We thank the reviewer for indicating this mistake which has been duly rectified.

545 cortical surfaces, attributed to active percussion (Profile C and E). The use of this tool for (546) percussion in (is) posterior to the previous phases of management 

OK thanks.

561 temporal order within the management phase: 1) a knapping phase and 2) a pounding 

OK

(562) phase (indicated in yellow and purple, respectively, Fig 11.). The figure uses the word ‘percussion’ rather than pounding, which implies hand-held striking. 

Pounding also implies a hand-held activity.

The marks on the rock’s angles could certainly be the result of several thrown impacts (training) 

The marks correspond to what we have observed by pounding activities in our own experiments with the same kind of limestone (Barsky et al., 2018).

…and are unlike the (what appears to be) pounding marks in Fig. 10. 

In Figure 10, the situated on the cortex marks correspond to percussion activity using the natural angle of the cobble. Marks observed on all the knapped edge with removals are posterior to the knapping. There is no rotation of the matrix during the percussion activity.

In Fig. 11, when this tool was used for percussion, the tool was rotated: the crests separating the facets are anthropically generated and the pounding activity is totally posterior: and situated on the crests (micro-removals around the edge similar to crushing observed on heavy-duty scrapers; see Barsky et al., 2018). Meanwhile, some double facetted breakage is testimony to crushing on a hard surface. Note that these are ALWAYS on the edge. These features exclude throwing as a possible source of these traces, as it would leave a more aleatory trace patterning.

The text states: (260) the limestone was used for different purposes; all mainly linked to percussive activities (261). A thrown rock is also percussive in that impact marks will show if it has been thrown and hit or landed on a hard surface. 

Repetition: see previous.

721 Bisi and colleagues [100] found on the surface at Ca 'Belvedere de Monte Poggiolo (Ca’ Belvedere)

OK

799 Sometimes the use of stone anvil attested (Fig 7)…. (Text missing ??)

OK

802 innovation [188]. This justifies some DOB sites being reaassigned Acheulian de la Torre 803 [137]. (to the as stated by?)

OK

Suggested Additional References

Wilson, A. D., Zhu, Q., Barham, L., Stanistreet, I., & Bingham, G. P. (2016). A dynamical analysis of the suitability of prehistoric spheroids from the cave of hearths as thrown projectiles. Nature, Scientific Reports, 6, 323.

Cannell, A. (2002). Throwing behaviour and the mass distribution of geological hand samples, hand grenades and olduvian manuports. Journal of Archaeological Science, 29, 335–339.

Cannell Alan (2018): Mass Distribution Analysis of Spheroid Manuports, Spheroid Artifacts, and the Lithics of Play Learning, Lithic Technology, DOI: 10.1080/01977261.2018.1460703

Espigares, M. P., Martínez-Navarro, B., Palmqvist, P., Ros-Montoya, S., Toro, I., Agustí, J., & Sala, R. (2013). Homo vs. Pachycrocuta: Earliest evidence of competition for an elephant carcass between scavengers at Fuente Nueva-3 (Orce, Spain). Quaternary International, 295, 113–125.

We have incorporated these thanks: however, we advise the reviewer that there is a new paper by Espigares et al. (2019) rather than the PhD, confirming the cut marks on bones of FN3 and BL. This paper has been integrated 

Response to Reviewer #2

The paper presents a new addition on the lithic assemblage from a well-known European Mode 1 site, Barranco Leon. Specifically, it focuses on a small subspheroid assemblage made on limestone that has been analyzed through a four-phase technological perspective. I was excited for reviewing this paper, but after finished I’m left with the feeling that more could have been done. Although the general data presentation is correct, I find a huge scarcity on some of the paper sections, especially on the one related to the Morphometric analysis, which to me should have been the biggest asset of the paper. 

To overcome this “scarcity”, we already have placed significant information in table 2:

- qualitative characteristics: alteration, general morphology;

- morphotechnical features: faciality, polarity, negative number;

- morphometrical quantitative data: size of each piece, weight, volume, as well as for the negatives and facets (average size, min. number/piece). 

However, to respond to this query we send additional Supporting Information file “S1 dataset” in which we present negative crest angles per tool and statistical data.

Therefore I recommend this paper to go through major revisions I strongly recommend a general English revision of the manuscript. Maybe the authors can ask an English native colleague to review it for them. 

This has been done.

In addition, here is a list of some of the writing mistakes I was able to find during my revision: 

We thank the reviewer and we have addressed all of these typos.

Lin.156: “we was apply” – OK changed to “we apply…”

Lin. 268: “to remedy to the difficulties” – Done. This phrase now reads “To overcome the difficulties…”

Lin. 176: “edorrheic” – Corrected “endorrheic”

Lin. 773: “be may” – Corrected to “…may even be…”

Lin. 447: “visibles” – Corrected to “visible” 

Lin. 613: “anges” – Corrected to read “angles”

Lin.799: “Sometimes the use of stone anvil attested” – Corrected to read: “That sometimes involved the use of a stone anvil…”

Lin. 835: “recongnized” – Done: “recognized”

INTRODUCTION 

Lin. 67: “… has been tentatively attributed to Kenyapithecus…” 

I’ve never seen the word Kenyapithecus published anywhere. Kenyanthropus platyopts is the name given to the species in its original paper and it’s also how appear in the several publications that mention it. To mention a couple: 

Leakey, M. G., Spoor, F., Brown, F. H., Gathogo, P. N., Kiarie, C., Leakey, L. N., & McDougall, I. (2001). New hominin genus from eastern Africa shows diverse middle Pliocene lineages. Nature, 410(6827), 433. 

Spoor, F., Leakey, M. G., & O'Higgins, P. (2016). Middle Pliocene hominin diversity: Australopithecus deyiremeda and Kenyanthropus platyops. Philosophical Transactions of the Royal Society B: Biological Sciences, 371(1698), 20150231. 

We thank the reviewer for the correction of this error; indeed the correct denomination is “Kenyanthropus platyops”: note to the reviewer that there is no ‘t’ at the end of the species name. Since there is no clear link established between toolmaking at the Lomekwi 3 site and this hominin we have decided to eliminate and have changed the phrase as follows: “…the Lomekwi 3 lithic assemblage (3.3 Ma, West Turkana, Kenya) predates the emergence of the genus Homo”.

Lin 78-79: “… generating a process of distinction from other primates, underlined by the development of technical skills.” 

It’s probably due to the way this sentence is written, but it looks that the authors are suggesting the mentioned process of distinction is somehow a conscious one. Furthermore, enough evidences of technology in chimpanzee contexts have been provided, so I would try to be careful with this kind of statements. 

Yes: this phrase was too simplified. We have added and modified this section for more clarity to read as follows:

“It therefore appears that more than one species of hominin – not all of the genus Homo -began to rely ever more significantly upon technologies, in an adaptive shift to limit constraints posed by the environment through object mediation. This change generated a process in which our hominin ancestors would come to distinguish themselves from other primates, by learning a comparatively high degree of technological skills.”

Lin. 87-91: “… Through time, the transition towards new reduction systems occurred within the OIC [28], progressively moving beyond the mechanics of the unifacial and unidirectional strategies, and leading to greater morpho-technological variability within some of the Oldowan tool kits, with orthogonal, unifacial discoid and multidirectional core management strategies…” 

This statement is something to take into consideration. Is the suggested transition a real transition? Advanced Oldowan reduction strategies can be found in OGS-7, Lokalalei 2C and Kanjera South, to mention some, and all those sites predate 2.0 Ma. The Oldowan manifests a large regional variability in its complexity, both synchronic and diachronic, so I find risky, to say the least, the suggestion of a lineal progressive evolution. It doesn’t seem to be supported by the current state of the art. I know the authors have quoted this paper later on the text, but I find it essential in order to work through this paragraph. 

Stout, D., Semaw, S., Rogers, M. J., & Cauche, D. (2010). Technological variation in the earliest Oldowan from Gona, Afar, Ethiopia. Journal of Human Evolution, 58(6), 474-491. 

Yes, in fact we do agree on this that this passage needed some, rather heavy editing, to be more comprehensible. First of all, we eliminated the beginning ‘Through time…” to efface the idea of linear evolution within OIC. We also removed the word “progressively” from the explanatory text, adding the following passage to remove ambiguities:

“While technical variability is observed within the OIC, its foundational features are largely uniform: small, non-retouched flakes, unidirectional or orthogonal core types, generally accompanied by a larger-sized pounding toolkit. However, Oldowan variability as described by Carbonell and colleagues [22], and Barsky [21], is attributable precisely to these same factors; most significantly, the different morphologies obtained using the unidirectional and orthogonal core reduction methods. Final core morphologies will vary in accordance to the raw materials, the length of knapping episodes and the cobble’s formal attributes. For classification purposes, these forms are attributed different denominations, while they in fact represent different stages in the application of these simple knapping systems (with little or no platform preparation). In some cases, new reduction systems did occur within the OIC [28], allowing hominins to move beyond the mechanics of the unifacial and unidirectional strategies, and to explore multidirectional core management strategies [29]”.

Thus, the phenomenon of Oldowan variability is illustrated, for example, in the novel core forms of the (referenced) Oldowan site of Fejej FJ-1 (Lumley and Beyene, 2004). We can also refer the reviewer to other papers discussing variability in the OIC:

In knapping strategies: 

Lumley, H. de, Barsky, D., Moncel, M.-H., Carbonell, E., Semaw, S., Cauche, D., Celiberti, V., Notter, O., Pleurdeau, D., Hong, M.Y., Rogers, M.J., 2018. First technical sequences in human evolution at East Gona (EG 10), Afar Region, Ethiopia. Antiquity 92 (365), 1151–1164. 

And secondary knapped flakes:

Barsky, D., Garcia, J., Martínez, K., Sala, R., Zaidner, Y., Carbonell, E., Toro-Moyano, I. Flake modification in European Early and Early-Middle Pleistocene stone tool assemblages. Quaternary International 316, 140-154. 

As new Oldowan sites are being discovered and systematically studied, it is becoming evident that the ideas originally proposed by M. Leakey- of stages of development within the Oldowan (however subtle they may be) are discernible. This paper focuses on one of those developments: the appearance of rounded (multidirectional) morphologies- focusing on this point of interest which is whether or not this formal attribute was intentionally sought out (implying relatively complex technological and cognitive functions), or whether it resulted from accidental percussive activities (as claimed by some specialists). The few pieces presented here from BL provide some answers, while the problematic is just starting to be opened up using modern technologies applied to stone tool analysis; in this case, 3D computational, diacritical and statistical analyses. We also discuss this interesting phenomenon by dealing with the significance of the appearance of PSSB in some Oldowan toolkits (or in some levels of Oldowan occurrences). Finally, the significance of our work is the scarcity of such forms in the European Oldowan. 

Lin. 118 – 121: “Based on these descriptions, the different categories comprising the PSSB can be distinguished from each other by direct observation in order to recognize deviances from the regular morphological-volumetric aspect of a sphere, developed around a central point (center of mass), which confers them a more or less rounded shape.” 

To me, that is just a way to recognize the inherent subjectivity of any non-quantitative lithic analysis. 

Indeed subjectivity is a major problematic in the recognition of PSSB in tookits worldwide. This is precisely why we are developing a more subjective methodology in order to deal with this difficulty. Up to now, numerous specialists have grouped a wide array of stone objects into the realm of spheroid morphologies and there is not always agreement on these choices. Even within a single site, as is the case here, it is difficult to avoid the ‘singling-out’ of some pieces using subjective criteria. However, the relative uniformity in knapping strategies at BL (and many other Oldowan sites, for that matter), make this task less arduous, since the multidirectional or polyhedron forms are scarcer. We argue that, in this paper, we avoid this problem by including two multipolar multifacial core forms into our study to test whether what we perceived ‘subjectively’ (that they were different from the pieces we had singled out as possible subspheroids) could be verified ‘objectively’ by the methodologies we develop here. We are satisfied with our results and intend, in future, to test our methodologies on other toolkits with, perhaps, more numerically important and similar kinds of implements. 

MATERIALS AND METHODS 

Lin. 258: “Cores are scarce and intensively reduced (3.1 % of the 1 562 pieces in flint)” 

How did the authors measure the reduction intensity of the cores? Which are the knapping strategies used? Do they correlate with the ones identified on the limestone assemblage? 

We thank the reviewer for this comment since we needed to specify here that this phrase refers to the flint cores. We have added specifications therefore to clarify between the flint and the limestone. We have also added and modified this paragraph as follows:

“Flint cores are scarce (3.1 % of the 1 562 pieces in flint) and intensively reduced compared to the limestone (11.2% of the 581 pieces in limestone). This greater intensity of reduction can be explained by the categorical differences in the assemblage concerning these two rock types: flint was used for obtaining small flakes, while limestone served mainly for percussive activities. It can also be explained by the relative scarcity of flint compared with limestone in the immediate vicinity of the site. Flint was collected in detrital position as small nodules, cobbles or plates, and the cores are very small. Contrastingly, the limestone was collected as cobbles of varying sizes and shapes [51, 52]; limestone cores are bigger than the flint ones and they often present reserved cortical surfaces. Bipolar-on-anvil stone reduction played an important role in the exploitation of both of these rock types; while freehand hard hammer methods are also recognized as significant [63]. Core reduction strategies, achieved by both of these methods, are described [63, 65] as recurrent and unifacial or orthogonal”.

The knapping strategies used for the limestone are: 

“Limestone cores are: unifacial (33.8% of the limestone cores) (types: unifacial unipolar, unifacial semi-peripheral and unifacial peripheral, unifacial centripetal, unifacial bipolar); bifacial (50.8% of the limestone cores) (types: bifacial bipolar, bifacial orthogonal with 2 and 3 directions of removals, bifacial multipolar); and multifacial (15.4% of the limestone cores) (type: multifacial multipolar).”

This has been added to the manuscript.

I feel like there is very few explanation regarding this topic, while a lot of information (maybe not that necessary) is provided on the general characteristics of the assemblage. 

We find that this comment is in contradiction with the previous one that asks for more general information about the assemblage. While the knapping strategies of BL are not the main focus of the present paper and have been largely published both in a monography and other English language scientific journals cited in our manuscript (see references: 21, 62, 63, 65, 66, 67), we have added relative frequency details of the different core types in accordance to our recent re-evaluation of the BL stone tool assemblage.

I also get that is not exactly the point of this paper to present the core features, but they are the elements that could possibly be correlated with the following PSSB discussion. 

This is an important point. The diacritical analysis we present here shows that a similar procedure of sequential, recurrent orthogonally-oriented knapping strategy leading to multifacial multipolar-type core forms is complimented: in the case of the sub-spheroids we identify here- by another phase of knapping. The latter (indicated in light blue on the diacritical drawings) is most interesting because the removals effectuated during this phase lend a rounded morphology to the pieces affected. Furthermore, the flakes resulting from this operation, effectuated after the latter knapping phase, were most certainly not the aim of the operation; as specified in the present manuscript:

“… the flakes attributed to this final phase were so thin that they would either have broken during extraction, or been useless for any cutting activity: this implies that the aim of this removals was not production. Rather, it seems that the exploitation was intended to add facets following the rounded morphology of the initial cobble”.

We argue that the diacritical analysis presented here provides sufficient comparative information separating the limestone polyhedral core forms (rare in the assemblage) from the sub-spheroids.

Lin. 265: “After reviewing the entire lithic collection, five limestone pieces…” 

The PSSB pieces sample is really small, five in total, a 0.2% of the whole lithic collection. I’m fully aware that there is nothing the authors can do about it; the sample is what it is, but they should be aware of the problem this sample poses if they intend to include the BL subspheroid assemblage into the large, pancontinental Oldowan discussion. 

Yes. “Small but significant” (to use an expression from Alperson-Afil & Goren-Inbar, 2016). We realize that we are presenting only a small component of the BL assemblage but, in spite of the numerous publications dedicated to this assemblage, it is the first time that a paper is specifically dedicated to this topic. In recognizing (ourselves but also other international researchers specialized in the Oldowan and the Acheulian), we immediately realized the significance of the presence of sub-spheroid morphologies in the oldest Oldowan in Europe. So, in spite of the small sample of pieces we are presenting here, we contend that it is a very important contribution to the scientific community.

Lin. 268-271: “To remedy to the difficulties relating to surface preservation of these five pieces and to recognize and verify their special features and assess the direction of their removal negatives, we decided to compare them with 2 of the best preserved multifacial multipolar cores (out of the 10 available in the collection)” 

I don’t fully understand why the authors did this. Establishing a comparison with multifacial multipolar cores implies that they thought about a relation between the two types of artifacts before the actual analysis of the PSSBs. If that is the case, I don’t understand why they limited the comparison to only two cores of the total assemblage, if taking all of them would have helped to enlarge the sample and obtain more reliable data. 

All the 10 multidirectional multifaceted cores have been carefully studied. But the 8 pieces not presented in this study report a state of alteration attributable to levels 3 and 4 (fig 3). However perceptible the directionality of the negatives, for the purpose of the material presentation we present the two pieces in which the impact points and direction of the negatives are extremely clear. It would be impossible to effectuate a detailed diacritical analysis on the rest of the pieces given their advanced state of alteration. We added this to the manuscript:

“In fact, only these two cores presented surface conservation sufficient to realize the diacritical study (visible impact points and removal directions)”.

These also relates with my previous comment on why the authors didn’t provide with a larger explanation on the core’s features. 

We have answered this query above.

RESULTS 

Lin.447-450: “For the pieces with visibles impact points (N=number of angle with impact point visible for each piece), we calculated the average of the knapping angles (kn.a): Tools D and E (134 °, N kn.a = 7 and 131 °, N kn.a = 8, respectively) showed more acute angles than for Tools F and G (121 °, N kn.a = 9 and 116 °, N kn.a = 4, respectively).” 

How can 134 and 131 degrees angles been more acute than 121 and 116 degrees angles? 

We really thank the reviewer for making us aware of the error. The term acute was corrected in the manuscript with the term obtuse. 

Lin. 451-453: “While the final morphologies of the tools occur as cuboid, polyhedral, polyhedral-rounded or rounded, the analysis of residual cortex indicates a priori anthropic selection of rounded cobbles” 

These statement necessarily implies that all the selected pieces are volumetrically reduced at the exact same stage, which they don’t know or at least haven’t explain anywhere on the paper. 

We have modified this section for more clarity as follows:

“It is important to underline that, while the final morphologies of the tools occur as cuboid, polyhedral, polyhedral-rounded or rounded, the analysis of the original shape of the cobbles, as assessed from the form of the surfaces displaying residual cortex, indicates a priori anthropic selection of rounded cobbles compared to the range of cobble forms available in the sedimentary context of the BL site [51, 52]”.

We now specify that, because there is a range of cobble forms available to the BL hominins, the observations made to the form of the cortical surfaces conserved on the pieces was key to understanding this interesting selection process.

Lin. 459-460: “The diachritical analysis allowed us to reconstruct each phase of manufacture for four of the tools (Tools D, E, F, G) (Fig 7, Fig 8, Fig 9, and Fig 10)” 

It might only be a matter of semantics, but I find that saying “each phase” might be a little bit risky. As they didn’t evaluate the reduction stage of the pieces, and assuming that new flake removals tend to erase the previous ones, I think it would be more accurate to say that they recognized the last phases of the sequence, not each one of them. 

OK. We have changed the phrase to: “The diacritical analysis allowed us to reconstruct the different phases involved in the manufacture…”

We also bring to the attention of the reviewer the methodological specifications explained in “stage 2) Diacritical study of the methodology section. Concerning the diacritical analysis, problematics such as the effacing of removals by other removals, are dealt with by assigning symbols to removals whose sequentiality cannot, for one reason or another, be precisely determined. Each of the different reasons for this (alteration, overlap, etc.), is identified using a different symbol and explained in detail in this section of the methodology. Despite this problematic, the ‘phase’ of knapping to which these surface blows belong is determinable. In addition, because the pieces presented here maintain relative proximity to the original form of the cobbles, as stated above, complete loss of knapping phases caused by very intense operative schemes is not a particular feature of these tools.

Lin. 467: “Tool D (Fig 7)…” 

I don’t really see how the yellow surface presented in Profile E correlate with the green one presented in profile B, so its sequential ordination seems unclear to me. 

We thank the reviewer for the particular attention given in the diacritical design that has allowed us to improve it and make the changes useful to make it more understandable.

We add here an image with relative magnification to answer to the reviewer and justify our change/improvement of figure 7.

As well visualized, the peripheral management phase (yellow) between the B and E profiles is separated from the cortical surface and from phase 3 (blue), which, as visible in profile D, separates the negatives series of phase 2 (in yellow). 

Even if, as indicated in the text (Lin. 489-500): “Phase 2 (indicated in yellow): some negatives forming the platform were orthogonally truncated as the piece was knapped along its periphery (Profiles B and E). Because they are not contiguous, it is impossible to determine which of these two profiles was knapped first”, there is no contiguity between the two faces, it is clear that the peripheral phase is subsequent the preparation of the platform (green phase 1) and previous the blue phase.

The impact point of negative 8 (profile E) has been corrected in new Figure 7 (and detailed for the Reviewer in an additional figure provided below Fig.rev1.). It uses negative 2 (phase 1) as a platform, just as the negative without number visible in profile F (phase 2) originates from the platform generated by negative 3 (phase 1). The connection between the two surfaces is visible in profile A. (Fig.rev1).

Even if, there is no contiguity between the B and E profiles (Fig. 7), the sequence between the phases 1 and 2 is appreciable, subsequently cut from phase 3. Therefore, the reading of the phases does not change. 

 Fig.rev1

Also, if a percussion platform is been created in Phase 1 (green), how the percussion platform for Phase 2 in Profile E appears to be the blue one (associated a Phase 3)? One possible explanation could be that the previous presence of Phase 1 removals were erased by the posterior Phase 3, but I cannot assure it just with pictures nor with the quality of the 3D scans. In any case, Figure 7 does not result very clear, and I recommend the authors to provide a more detailed explanation of the relationship between surfaces and Phases. 

As indicated by the figure caption Fig 7, Lin. 514-515: “Phase 3 (blue): Series of removals from a cortical platform (Profile D), cutting the structure transversely (Phase 2) and frontally (Phase 1).”

More detailed explanation of the relationship between surfaces and Phases is explained in the previous answer. 

Lin. 492: “Tool E (Fig 8)…” 

In Profile A, why aren’t the Phase 1 removals numbered? I don’t see any identification problem and I believe are the same ones marked on Profile F. 

Exactly. They are the same ones marked on Profile F. Numbers have been added (new Figure 8).

Lin. 511: “Tool F (Fig 9)…” 

In general, not only on this tool, I think the graphic design on the diachritical schemes should be improved, as sometimes is leading to misinterpretations. For example, in Profile E, Phase 2b (dark yellow), the scar marked as 6* is clearly cutting into the one marked as 8*. Therefore, one would say that scar is posterior to 8* and not a previous one, when, judging by the actual photo and the authors explanation that is not the case. 

We have made appropriate corrections to reduce the misinterpretation. We thank the reviewer for these comments allowing us to improve the presentation of the images.

 In this case no: the scar marked as 6* is NOT clearly cutting into the one marked as 8* as the reviewer states, but: 8* is clearly posterior to 6*. The numbering is therefore correct.

Lin. 643-645: “This characteristic of the second group gives the objects a more rounded shape than those of the first group, which, contrastingly, tends to present more angular and irregular profiles”. 

That is a little bit obvious, isn’t it? 

Yes, perhaps, but as the reviewer has stated, this methodology allows us to confirm subjective observations and transform them into objective data.

I really appreciate the intention of been thorough and pursue a quantitative approach with this paper, but the statement that a spherical volume has less acute angles than a polyhedron is hardly a result. 

It is a result: thanks to the diacritical analysis that shows that a final phase of removals- not related to debitage (since the flakes would have been either too thin or broken) -that gives the spherical shape to the pieces, was effectuated intentionally in the aim of conferring a rounded form: this is highly significant in the Oldowan context and is at the crux of the questions surrounding spheroids in the literature since the 1970’s. It was important for us to test our observations using an objective methodology.

In general terms, the morphometric analysis here presented looks a little bit scarce. These section of the paper is crucial to me and should be extended. 

This work is not based solely on morphometric analysis (to which in any case we have integrated Suplementary information (S1 Dataset)), but there are several issues of fundamental importance presented in this paper , as: 1) we documented for the first time subspheroids in the European late Oldowan; 2) we propose a methodology using 3D analysis for studying subspheroids and discriminate them objectively from similar polyhedron morphology; 3) we report the amplitude and variability of removal angles determinant for PSSB categories; furthermore 4) the diacritic analysis reveals preconceived template in the subspheroids from Barranco León, underling a 5) cognitive evolution within the Oldowan technocomplex.

In this paper, we presented information that can be used for comparative study of other collections that include PSSB morphologies. Definitely, considering the interest in the topic, a deeper study and morphometric analysis can be carried out, on a collection that presents a greater number of spheroid morphotypes. A larger sample will allow us to further test our new methodology and to obtain more results that can then be compared with the data from BL. This does not underestimate, as mentioned above, the significance of presenting this small collection to the scientific community. Only five pieces, but of fundamental importance in cognitive terms and despite its quantitative scarcity documents for the first time subspheroids in the European late Oldowan.

DISCUSSION 

Lin. 709: “When the PSSB are accompanied by Acheulian elements in an assemblage, their cultural attribution is moved from Oldowan to Acheulian (Table 4)”. 

Well, of course. It is a set of artifacts displaying diachronic presence. What is the point of remarking the artificial nature of any taxonomic category? 

Because we use these categories as the foundation of chrono-cultural attributions: whether ideal or not, this remains the dominant methodology used to define prehistoric cultures even today. It is important to remark the artificial nature of these categories so that we can refine them and recognize that the paradigm tool-type=culture is far from clear-cut. This is one of the main problematics sparking discussions around the significance of spheroids (and bifaces, for that matter), ever since they were suspected to be intentional tool forms in Oldowan African toolkits. We note that they are absent from the oldest African assemblages, such as Kada Gona and Ounda Gona, AL 666 and Lokalelei… does this mean that their ‘diachronic presence’ is unidirectional?

Especially when those categories doesn’t take in account actual technological features, complexity parameters or quantitative data, but presence/absence of recognizable tools. 

But here we are taking into account these parameters.

Lin. 718-719: “Contrastingly, in sites attributed to the Oldowan, PSSB are associated with percussive implements and there are no handaxes”. This is, again, an example of circular reasoning that doesn’t say anything about the actual PSSBs. 

We have changed the phrase for clarity as follows:

“…in sites without handaxes attributed to the Oldowan, PSSB are associated with percussive tools…”

The statement is linked to the state-of-the-art in defining the denomination problematic that is central to the Oldowan-Acheulian debate: to define internal variability within one or another of these ‘cultural umbrella’ terms, or to distinguish specific features with which to distinguish between them. It serves here to explain the categories presented in Table 4. What it says about PSSBs is simply that, as Table 4 demonstrates, their general attribution to the Oldowan rather than to the Acheulian is based on their association-or-not with the emblematic Acheulian tools, which are handaxes. Arguably, the fact that they are generally associated with percussive tools gives fuel to the hypothesis that they may somehow be related to percussive activies, grosso modo. 

Lin. 745-747: “The ability of hominins to pursue a recurrent strategy in the management of multifacial multipolar cores, as in the Orce subspherods described here, represents a step forward compared to the older assemblages in terms of cognitive advancement”. 

There are older chronologies with long, systematized knapping sequences (See again Lokalalei 2C or even Gona). 

We are familiar with these and have largely discussed them, for example in 22, 28 and 142, and even more recently in:

Lumley, H. de, Barsky, D., Moncel, M.-H., Carbonell, E., Semaw, S., Cauche, D., Celiberti, V., Notter, O., Pleurdeau, D., Hong, M.Y., Rogers, M.J., 2018. First technical sequences in human evolution at East Gona (EG 10), Afar Region, Ethiopia. Antiquity 92 (365), 1151–1164. 

 Once again, these systematized and relatively long knapping sequences recognized in these ancient assemblages are still based in the unidirectional recurrent and orthogonal recurrent knapping strategies (although Lokalelei does show some preparative stages to the knapping like split cobble platform preparation). But these are NOT configuration- nor do the indicate that hominins were seeking to give an intentional shape to their core-forms. 

I don’t really see the step forward if we are referring to a set of lithic artifacts largely heterogenic that doesn’t present clear attributes of a final morphology. 

In light of our previous comments, any intentional shaping; be it in the heavy-duty or in the light-duty toolkits attributed to the Oldowan, is highly significant in terms of taking a ‘step forward’ in the sense of cognitive evolution. None of the oldest Oldowan toolkits show any signs of advancement towards purposefully obtaining a pre-defined final morphology. We argue that our diacritical analysis, although limited to only a few pieces, is significant, in that it constitutes a subtle indication of this ‘step forward’ within a clearly Oldowan context of Europe.

I find difficult to refer “PSSB” as a morphotype. It is a term that widely refer to diversity, but the elements that comprise them are quite different. So far, I’ve not seen any attempt to explain why that variability appear on the archaeological record.

We do not refer to PSSB as ‘a morphotype’: please refer to the definitions in the introductory sections of the manuscript.

We propose that this paper provides an effort to explain variability- at BL –at least between the polyhedron and subspheroid ‘morphotypes’. We also provide ample bibliography. Obviously, we cannot claim to explain the variability within the PSSB group in the archeological record.

Lin. 781-784: “We define polyhedron morphologies by higher variability in their angles (Fig 13), which are also more acute, likening them to the group of multifacial cores. Comparatively, truly subspheroid morphologies display more open facet angles and greater uniformity”. 

Again, this statement relates with my previous comment on subspheroids being more rounded than polyhedrons on the Results section.

Relating in the same way to the reviewer’s previous comment, we reiterate that, one of the central contributions of our manuscript is: that it provides a methodology which we hope, in future, will be more widely applied to toolkits containing PSSBs, to assist researchers in avoiding subjectivity in defining pieces that belong to one or another of the morphotypes attributed to this group: polyhedrons, subspheroids, spheroids or bolas.

On behalf of myself and all of the contributing authors, we sincerely thank the reviewer for the constructive criticisms and we believe that we have provided adequate answers to each of his/her queries. We contend that this review- and the modifications to our original manuscript that it entails –has greatly improved the clarity of our arguments and the general readability of this paper.

---

## [Decision Letter · Decision Letter 1]

27 Nov 2019

PONE-D-19-22634R1

Subspheroids in the lithic assemblage of Barranco León (Spain): recognizing the late Oldowan in Europe

PLOS ONE

Dear Mrs. Titton,

Thank you for submitting your manuscript to PLOS ONE. After careful consideration, we feel that it has merit but does not fully meet PLOS ONE’s publication criteria as it currently stands. Therefore, we invite you to submit a revised version of the manuscript that addresses the points raised during the review process.

I appreciate the improvements made to the paper. I obtained the views of one of the original reviewers, and relatively minor issues arose in the Results and Discussion sections. I request that you assess these points, which seem to be reasonable. 

We would appreciate receiving your revised manuscript by Jan 11 2020 11:59PM. To enhance the reproducibility of your results, we recommend that if applicable you deposit your laboratory protocols in protocols.io, where a protocol can be assigned its own identifier (DOI) such that it can be cited independently in the future. For instructions see: http://journals.plos.org/plosone/s/submission-guidelines#loc-laboratory-protocols

We look forward to receiving your revised manuscript.

Kind regards,

Michael D. Petraglia, Ph.D.

Academic Editor

PLOS ONE

Reviewers' comments:

Reviewer's Responses to Questions

**Comments to the Author**

1. If the authors have adequately addressed your comments raised in a previous round of review and you feel that this manuscript is now acceptable for publication, you may indicate that here to bypass the “Comments to the Author” section, enter your conflict of interest statement in the “Confidential to Editor” section, and submit your "Accept" recommendation.

Reviewer #2: All comments have been addressed

2. Is the manuscript technically sound, and do the data support the conclusions?

Reviewer #2: Yes

3. Has the statistical analysis been performed appropriately and rigorously? 

Reviewer #2: Yes

4. Have the authors made all data underlying the findings in their manuscript fully available?

Reviewer #2: Yes

5. Is the manuscript presented in an intelligible fashion and written in standard English?

Reviewer #2: Yes

6. Review Comments to the Author

Reviewer #2: The authors have successfully resolved most of the issues highlighted in the previous revision. I really appreciate the effort made on the manuscript edition, and the Introduction, Context of BL, Materials and Methods sections are relatively clear to my view. I also consider very positive the inclusion of the Supporting Information File in the morphometric analysis.

However, the Results section remains a little problematic to me. As I mentioned in the previous revision, the diacritical schemes presented are not as clear as they should be, and could easily be misinterpreted, probably due to the type of drawing. The authors have made minor changes in Figures 7 and 8, but I still feel that more effort could be made regarding the informative content and aesthetics of the Figures. There is no disagreement with the authors on the way the phases have been identified; just the way this information has been presented.

I would also like to discuss a bit more the relationship between the subspheroid sample analysed and the multifacial cores. Furthermore, I suggest the authors to provide a deeper explanation about why they chose to compare the PSSBs in the first place. During the whole manuscript, the two multifacial multipolar cores have been considered as part of the PSSBs assemblage, or at least that is the impression I got, despite this paragraph included in the Materials and Methods section in which the authors say that are different things:

“After reviewing the entire lithic collection, five limestone pieces (0.2 % of the whole 294 collection) were classified by their morphological characteristics and special technological features as attributable to the PSSB group as defined by Kleindienst [35], Leakey [13] and 296 more recently by Tixier & Roche [34]. To overcome the difficulties relating to surface preservation of these pieces and to recognize and verify their special features and assess the direction of their removal negatives, we decided to compare them with 2/10 of the multifacial multipolar cores. In fact, only these two cores presented surface conservation sufficient to realize a diacritical study (visible impact points and removal directions)”.

I think this is a key point on the paper, specially taking in account that the diacritical schemes have been fully performed in 2 of the 5 subspheroids (D and E) and in the 2 multifacial multipolar cores (F and G). The recognition of the same management phases in artefacts that have been classified as different is worth detailed explanation and further comment.

In the Discussion section, the authors comment: “the diachritic analysis has allowed us to evaluate that, at least for the BL site, sub-spheroid morphologies result from a well-reasoned, organized shaping process that sometimes involved the use of a stone anvil (Fig 7). Meanwhile, such preconceived morphological templates transposed onto stone are usually attributed to the Acheulian, concretized, for Isaac [191], by the presence of innovative handaxes and standardized tools with recurring morphologies”. Again, I suggest being careful with this kind of statements, and this is the reason: the diacritical analysis performed in Tools D and E show what appears to be a shaping phase, supporting the author’s statement. I cannot really value Tool C, as its diacritical scheme is partial. Also, I’m not including the results coming from Tools F and G as the previous paragraph strictly refers to “sub-spheroid morphologies” and, as I mentioned, the relation of the multifacial multipolar cores with the PPSBs remains a bit confusing to me. The other two PPSBs (Tools A and B) were discarded from the analysis due to a “high degree of alteration” (Lin. 504).

In the end, from the initial 5 PPSBs and 2 cores included in the original sample, only two of them (Tools D and E) show evidence of shaping. Again, I am not disagreeing with the authors; I just want to clarify why I think that the general interpretation of these results has to be cautious. I think the original sample is limited and extrapolations made from two artefacts can be problematic.

I do not consider the things I have pointed out as Major Revisions. There are very specific issues that are easily solvable. However, I consider necessary feedback in each one of them before seeing this paper successfully published.

Again, I want to thank the authors for the time and effort they put in correcting the previous version of this manuscript.

7. PLOS authors have the option to publish the peer review history of their article (what does this mean?). If published, this will include your full peer review and any attached files.

Reviewer #2: No

---

## [Author Response · Author response to Decision Letter 1]

4 Jan 2020

Answers to Reviewer #2 

The authors have successfully resolved most of the issues highlighted in the previous revision. I really appreciate the effort made on the manuscript edition, and the Introduction, Context of BL, Materials and Methods sections are relatively clear to my view. I also consider very positive the inclusion of the Supporting Information File in the morphometric analysis. 

However, the Results section remains a little problematic to me. As I mentioned in the previous revision, the diacritical schemes presented are not as clear as they should be, and could easily be misinterpreted, probably due to the type of drawing. The authors have made minor changes in Figures 7 and 8, but I still feel that more effort could be made regarding the informative content and aesthetics of the Figures. There is no disagreement with the authors on the way the phases have been identified; just the way this information has been presented. 

We agree with the Reviewer here and we are very grateful for this remark as it has allowed us to make a far better visual presentation of our schemes. The pieces have been redesigned improving their graphic representation. 

I would also like to discuss a bit more the relationship between the subspheroid sample analysed and the multifacial cores. Furthermore, I suggest the authors to provide a deeper explanation about why they chose to compare the PSSBs in the first place. 

To better explain and emphasize the concept we added in the manuscript the sentence underlined in yellow.

During the whole manuscript, the two multifacial multipolar cores have been considered as part of the PSSBs assemblage, or at least that is the impression I got, despite this paragraph included in the Materials and Methods section in which the authors say that are different things.

In answer to ‘why’ we chose to compare PSSBs in the first place: 

1) We had an assemblage containing pieces that were evidently reduced using the same strategy: multifacial multipolar.

2) Some of these pieces presented a spheroid morphology (i.e. they were more rounded).

3) We needed, therefore to elaborate a scientific and objective methodology in order to check our determination that these pieces were really different from the multipolar multifacial cores. 

To further clarify this in the text selected by the reviewer, we added the highlighted phrases indicated as follows:

“After reviewing the entire lithic collection, five limestone pieces (0.2 % of the whole 294 collection) were classified by their morphological characteristics and special technological features as attributable to the PSSB group as defined by Kleindienst [35], Leakey [13] and 296 more recently by Tixier & Roche [34]. We compare the pieces that share the multifacial and multidirectional management strategies, including those with semi-rounded/rounded morphologies (the PSSB), using diacritical analysis. In order to evaluate whether or not the PSSB can be isolated with respect to the cores, a morphometric analysis and statistical was carried out. To overcome the difficulties relating to surface preservation of these pieces and to recognize and verify their special features and assess the direction of their removal negatives, we decided to compare them with 2/10 of the multifacial multipolar cores. In fact, only these two cores presented surface conservation sufficient to realize a diacritical study (visible impact points and removal directions)”.

Cores are not defined as PSSB, but are included in the text and in figure 2 as a comparative sample. But, in order to eliminate the subjectivity in the subdivision of categories, during the analysis, the sample was evaluated as a whole. In fact, the 7 pieces are described in the results section as: “...multipolar and multifaceted limestone tools...” all of them presenting this characteristic (but distinguishable by their different morphologies). In the analysis: surface alteration; dimensional features (here we further underlined in the Table 2: Tools F and G are polyhedral cores used for comparative purposes); diacritical study, percussion marks analysis, and in the morphometric analysis as well, the pieces are all named as tools and not defined or included in a PSSB type assemblage. 

The statistical/morphometric analysis allows us to re-divide the sample into two groups (A and B): “Although the Lubishew’s test indicates only a moderate degree of discrimination between groups, descriptive statistics highlights a morphological distinction between the forms, linked to the width of the angles.”

Group A is different from B (spheroid and subspheroid morphologies) and is characterized by tools with more acute angles with higher variability in their amplitude (Table 3 and Figure 13), in which we can find multifaceted and multipolar cores (with irregular morphology) and the polyhedral PSSB subcategory.

“In this study, the differentiation between polyhedrons, subspheroids and spheroids is not, therefore, based on criteria relating to raw materials, but rather has been made using the width of the angles separating the facets of each piece. We define polyhedron morphologies by higher variability in their angles (Fig 13), which are also more acute, likening them to the group of multifacial cores. Comparatively, truly subspheroid morphologies display more open facet angles and greater uniformity.”

We therefore take into account throughout the manuscript, even if all the pieces are included in the same analysis, the distinction between the PSSB and multipolar cores. As well as the subcategories PSSB: polyhedron, spheroids and subspheroids.

We hope therefore, that the sentences added in the text (indicated in yellow) will make the concept clearer.

I think this is a key point on the paper, specially taking in account that the diacritical schemes have been fully performed in 2 of the 5 subspheroids (D and E) and in the 2 multifacial multipolar cores (F and G). The recognition of the same management phases in artefacts that have been classified as different is worth detailed explanation and further comment. 

In the Discussion section, the authors comment: “the diachritic analysis has allowed us to evaluate that, at least for the BL site, sub-spheroid morphologies result from a well-reasoned, organized shaping process that sometimes involved the use of a stone anvil (Fig 7). Meanwhile, such preconceived morphological templates transposed onto stone are usually attributed to the Acheulian, concretized, for Isaac [191], by the presence of innovative handaxes and standardized tools with recurring morphologies”. Again, I suggest being careful with this kind of statements, and this is the reason: the diacritical analysis performed in Tools D and E show what appears to be a shaping phase, supporting the author’s statement. I cannot really value Tool C, as its diacritical scheme is partial. Also, I’m not including the results coming from Tools F and G as the previous paragraph strictly refers to “sub-spheroid morphologies” and, as I mentioned, the relation of the multifacial multipolar cores with the PPSBs remains a bit confusing to me. The other two PPSBs (Tools A and B) were discarded from the analysis due to a “high degree of alteration” (Lin. 504). 

Beyond the discussion we have already described similarities as differences in management phases in artifacts that have been classified as different: section Results (lines 619-652) and with figure 12. But, to fill this doubt, we decided to integrate in the discussion the following sentence:

The concept of acquisition and transmission reflected in any systematic production of forms is qualified as culture. Even though the creation of a platform and subsequent peripheral management of a surface may seem simplistic, as in the case of orthogonal knapping, it became emblematic of Oldowan culture [135]. The subspheroid morphologies and the multidirectional cores of BL, for their repetition in the first phases of the management, can’t be fit in the classic opportunistic flake production. Furthermore, we have to take in account that, for the site under consideration, not only the two subspheroids (Tools D and E) show evidence of shaping, because other intentionally repeated morphotypes are present: as for the case of heavy-duty scrapers which have been defined also by intentional shaping [135]. This feature becomes more characteristic from the Acheulian techno-complex.

In the end, from the initial 5 PPSBs and 2 cores included in the original sample, only two of them (Tools D and E) show evidence of shaping. Again, I am not disagreeing with the authors; I just want to clarify why I think that the general interpretation of these results has to be cautious. I think the original sample is limited and extrapolations made from two artefacts can be problematic.

In answer to this query: we believe that although we have a limited sample, the presence of morphologies with repeated manufacture allows us to state, while using all the necessary caution, that there is a mental preconception in this hominin cultural group. Looking to other publications on the Oldowan, such preconceived morphotypes are always present in limited numbers: retouched denticulate tools, heavy-duty scrapers or ‘loosely configured tools’ (51). For example, in BL there are only 7 heavy-duty scrapers and these tools are also very scarce in other Oldowan sites where they have been identified. We argue that this is really one of the main characteristics of the Oldowan that changes later in the Acheulian, when recognizable standardization: real ‘tools’ are more numerically represented and therefore more easily discernible. Although there are only 2 pieces with optimal preservation that allow us to fit them perfectly into the subspheroid morphotype, we cannot ignore their presence.

I do not consider the things I have pointed out as Major Revisions. There are very specific issues that are easily solvable. However, I consider necessary feedback in each one of them before seeing this paper successfully published. 

We hope to have been clear in our answers, and to have resolved any remaining doubts on behalf of the reviewer.

Again, I want to thank the authors for the time and effort they put in correcting the previous version of this manuscript. 

We thank the Reviewer for his active contribution to the improvement of our work.

---

## [Editor Report · Decision Letter 2]

13 Jan 2020

Subspheroids in the lithic assemblage of Barranco León (Spain): recognizing the late Oldowan in Europe

PONE-D-19-22634R2

Dear Dr. Titton,

We are pleased to inform you that your manuscript has been judged scientifically suitable for publication and will be formally accepted for publication once it complies with all outstanding technical requirements.

With kind regards,

Michael D. Petraglia, Ph.D.

Academic Editor

PLOS ONE
---

## [Editor Report · Acceptance letter]

23 Jan 2020

PONE-D-19-22634R2 

Subspheroids in the lithic assemblage of Barranco León (Spain): recognizing the late Oldowan in Europe 

Dear Dr. Titton:

I am pleased to inform you that your manuscript has been deemed suitable for publication in PLOS ONE. Congratulations! Your manuscript is now with our production department. 

With kind regards,

on behalf of

Professor Michael D. Petraglia 

Academic Editor

PLOS ONE